# DMRG study of strongly interacting $\mathbb{Z}_2$ flatbands:
# a toy model inspired by twisted bilayer graphene

P. Myles Eugenio[1, 2, *] and Ceren B. Dağ[3, 1]

[1]*National High Magnetic Field Laboratory, Tallahassee, Florida, 32304, USA*
[2]*Department of Physics, Florida State University, Tallahassee, Florida 32306, USA*
[3]*Department of Physics, University of Michigan, Ann Arbor, Michigan 48109, USA*

Strong interactions between electrons occupying bands of opposite (or like) topological quantum numbers (Chern= $\pm 1$), and with flat dispersion, are studied by using lowest Landau level (LLL) wavefunctions. More precisely, we determine the ground states for two scenarios at half-filling: (i) LLL's with opposite sign of magnetic field, and therefore opposite Chern number; and (ii) LLL's with the same magnetic field. In the first scenario – which we argue to be a toy model inspired by the chirally symmetric continuum model for twisted bilayer graphene – the opposite Chern LLL's are Kramer pairs, and thus there exists time-reversal symmetry ($\mathbb{Z}_2$). Turning on repulsive interactions drives the system to spontaneously break time-reversal symmetry – a quantum anomalous Hall state described by one particle per LLL orbital, either all positive Chern $|++\cdots+\rangle$ or all negative $|--\cdots-\rangle$. If instead, interactions are taken between electrons of like-Chern number, the ground state is an $SU(2)$ ferromagnet, with total spin pointing along an arbitrary direction, as with the $\nu = 1$ spin-$\frac{1}{2}$ quantum Hall ferromagnet. The ground states and some of their excitations for both of these scenarios are argued analytically, and further complimented by density matrix renormalization group (DMRG) and exact diagonalization.

## I. INTRODUCTION

The unprecedented nature of the phenomena exhibited by twisted bilayer graphene (TBG) devices at various electron densities about charge neutrality (CN), including correlated insulated states [1–3] and superconductivity [4, 5], has opened the door to many experimental [1–7] and theoretical [8–26] studies since their initial discovery by *Y. Cao, et al* [1, 4] in 2018. Unlike band insulators, which occur when the Fermi energy falls in the gap between bands of a periodic crystal, fully occupying the lower band with electrons, these correlated insulating states are a result of (and cannot be described without) interactions, and can occur at any integer filling of a band. The strong repulsive interactions allow the electronic energy to be minimized by separating the particles as far apart as possible, producing a localized state necessary for insulation [27, 28]. Of principle importance to such an interaction dominated paradigm is the presence of flat or nearly flat bands [29], with bandwidths smaller than the interaction energy. In TBG, and similarly other Moire heterostructures (MHS) [30–34], such nearly flat bands are the result of an emergent long-range periodic potential. The $l_M \sim 13$nm periodic potential in TBG forms out of the overlap between two stacked single-layer honeycomb lattices, relatively twisted at magic angles [8, 12, 30, 31]. Multiple theoretical estimates place the bandwidth in TBG to be no larger than 10 meV [8–10], smaller than the ∼23 meV Coulomb energy observed by Scanning Tunnelling Microscopy (STM) [6, 7].

While the above prescription for an insulator – minimization of the interaction energy by localizing the elec-

trons – seems straightforward, it is complicated by the topologically non-trivial nature of the nearly flat bands [10, 12, 30]. In two dimensions, a complete orthonormal set of single-particle states, which are exponentially localized in both directions, cannot be constructed from bands carrying non-zero Chern number [10, 12, 19, 30]. Roughly speaking, one can understand this non-trivial topology in analogy with the quantum Hall (QH) [35–42], in which the Landau levels have zero dispersion and Chern number $\pm 1$, depending on the sign of the magnetic field. As such, there exists a gauge similar to the Landau gauge, which preserves localization in one direction, albeit sacrificing localization in the other, i.e the hybrid Wannier states [43–46].

However, while topological, the total Chern number of the narrow bands in TBG is zero. This can be understood within the continuum model of TBG [8], where the like-valley valence and conduction bands carry opposite Chern number $C = \pm 1$, and are guaranteed to touch by the combination of a $180^o$ rotation and time-reversal ($C_{2z}T$) [8, 9, 12, 13, 15, 17]. Yet unlike a topologically trivial system, it is possible to construct pairs of hybrid Wannier states which carry opposite Chern numbers, and which map into each other under $C_{2z}T$ [18]. If we consider again the QH, this would amount to having both the LLL and its $C_{2z}T$-partner, the latter being just the same Landau level wavefunction with opposite sign of the magnetic field, and which is designated by the group $\mathbb{Z}_2$.

In order to make such an analogy with the QH more precise, let us consider the limit of the chirally symmetric continuum model (CSCM) [15], where the narrow bands within the same valley are everywhere degenerate and *exactly* flat. Special to this limit, opposite Chern bands within the same valley are graphene sublattice polarized [15, 18]. This suggests that a sublattice anisotropy (i.e breaking $C_{2z}$ [2, 3, 16]) would split the intra-valley de-

* eugenio@magnet.fsu.edu

generacy into two separated flat bands of opposite Chern number, each with a degenerate time-reversal partner living in the other valley. Depending on the size of their gap $\Delta$ relative to the strength of the interaction, two such scenarios emerge in which $\mathbb{Z}_2$ LLL's as an analogy is inspired: (1) If the gap $\Delta$ is larger than the interaction strength, and thus mixing between the conduction and valence bands is suppressed. Despite the broken $C_{2z}T$ symmetry, the time-reversal symmetry guarantees that bands of opposite valleys are Kramer pairs [16]. Or (2), the chiral limit with $\Delta = 0$, where the conduction and valence bands are flat and degenerate everywhere. And thus Kramer pairs exists both between and within a valley.

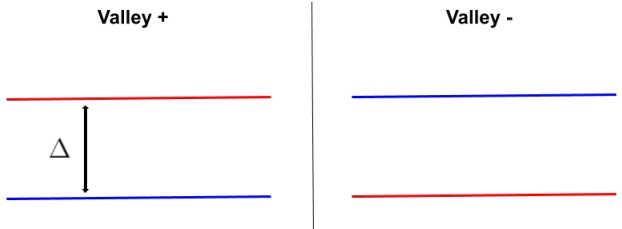

FIG. 1. Two flat Chern bands per valley: red (blue) bands are Chern $+1$ $(-1)$. The exact order depending on the sign of $\Delta$. Time-reversal symmetry guarantees valley degeneracy even if $C_{2z}$ is broken.

Not counting the spin degree of freedom, the total degeneracy of Scenario (1) and (2) is two and four respectively. The larger degeneracy in the latter brings with it more integer fillings and possible phases; however, for the purposes of this study, we focus on Scenario (2) and assume valley polarization, such that there is only a single intra-valley Kramer pair. In this limit, Scenario (1) and (2) have the same degeneracy structure, both describing an interacting system of fermions occupying opposite-Chern bands. It was previously pointed by out by *Bultinck, Chatterjee, and Zaletel* [16] that such a system resembles a bilayer QH problem [41, 42] with one flux quantum per unit cell [16], except with opposite layers experience opposite-sign magnetic fields.

In the original bilayer QH problem, a small anisotropy of the inter-layer repulsive interaction appears as an attraction between electrons and holes in opposite layers, and consequently the system is unstable towards the formation of an inter-layer coherent state [16, 41, 42]. As such, one might be led to believe that strong interactions between $\mathbb{Z}_2$ LLL's would too lead to a Chern zero state; however, as was previously explored with mean field theory [16], and we show here both analytically and numerically, this is not the case. The ground state is Chern-polarized and stable.

In this paper, we construct the toy model for strongly interacting opposite-Chern flat bands using continuum LLL wavefunctions. We focus our study on $\nu = 1$ electrons per flux quantum, and ask: (i) Is the ground state a Chern polarized (quantum anomalous Hall) state? And

(ii) is such a state stable, unlike the bilayer QH problem? We argue analytically that in the presence of repulsive interactions, the system will choose a Chern insulator, owing to the decomposition of the interaction into a sum of positive semi-definite terms [10, 17], which always favours one flavor of electron per single-particle orbital. Because of the $\mathbb{Z}_2$ symmetry, there are two such Chern polarized states in the ground state, which we denote: $|++\cdots+\rangle$ and $|--\cdots-\rangle$.

Additionally, we study the case involving distinct fermions with identical Chern number, but distinguished by an introduced pseudospin $\{\uparrow, \downarrow\}$. In the absence of any interaction anisotropy, this system is identical to the well studied $\nu = 1$ spin-$\frac{1}{2}$ QH ferromagnet [35, 39, 41]. We show the ground state is spin-polarized with one particle per LLL orbital, i.e $|\uparrow\uparrow\cdots\uparrow\rangle$, in addition to a degenerate manifold of states, extensive in the system size, connected by $SU(2)$ rotations. Opposite pseudospins may be taken to represent opposite layers of the bilayer QH problem [41, 42], where both layers experience the *same* sign of the magnetic field. Thus the bilayer QH problem would be unstable against an inter-layer anistropy, which would break the $SU(2)$ symmetry and split the extensively degenerate manifold in favour of a layer-coherent state. In comparison, the ground state manifold of the $\mathbb{Z}_2$ QH problem is doubly degenerate, independent of the system size. Therefore, as long as a gap separates its ground state manifold from the higher excited states, the $\mathbb{Z}_2$ system would be stable.

In order to confirm the existence of a gap, we utilize DMRG [17, 47–54] and exact diagonalization (ED) [17]. This is done by rolling our system into a cylinder, and utilizing the Landau gauge [37] to project our 2D continuum interactions into 1D discrete pseudopotentials [38]. Such a mapping is possible because of the topological non-trivial nature of the LLL's [47–51], and allows us to think of the LLL orbitals as slicing the cylinder into a 1D chain of "sites", with each site corresponding to the center of a LLL wavefunction (Fig 2).

DMRG reveals a consistent gap between ground and first excited states for all systems with various ratio of the cylinder circumference to the magnetic length $L_y/l_B = 8, 10, 12, 15, 20$ at large enough total orbital number $N$ in the $\mathbb{Z}_2$ case and where the Coulomb screening length is $l_s = 1 \ l_B$. We find that the gap shrinks with increasing screening length, crashing for sufficiently large $l_s$, beyond which the one-particle-per-site order melts into a state with particle density largest at the edge and center. Before the gap vanishes, the excited state of the $\mathbb{Z}_2$ Hamiltonian with open boundary conditions are edge states described by a change of $-2$ in the total Chern number of the ground states.

For the spin-$\frac{1}{2}$ QH ferromagnet, the gap computed by DMRG shrinks as $N$ increases for all $L_y/l_B$. The excited states converge to having one-particle-per-site density with increasing $N$, and are irreducible representations of the $SU(2)$ symmetry, but with a total spin $-2$ less than the total spin of ground states.

Because of the simplicity afforded by such an advantageous choice of basis, our $\mathbb{Z}_2$ model allows us to study the problem of strongly interacting opposite Chern bands in systems with larger cylinder circumferences than has been done previously by using tight-binding models, such as in Ref [17], which was limited to $L_y = 3$ unit cells. Even though we study larger circumference cylinders, the Matrix Product State (MPS) bond dimension of the ground state is many orders of magnitude smaller than Ref [17]. In fact, despite being in the limit where the ratio of bandwidth-to-interaction-strength is zero, i.e the extremely strong coupling limit, the bond dimension of our ground states remains 1; as opposed to Ref [17], where the bond dimension grows with the inverse of that ratio.

If we were to introduce a periodic potential, and therefore induce a bandwidth, the ground state bond dimension could only grow, a behavior which is opposite and hence complementary to the bond dimension scaling of Ref [17]. Nonetheless, our model successfully captures the spontaneously broken fully polarized ground state without introducing a finite bandwidth, thus highlighting the importance of the topology in the ground state physics. Introducing a small but finite bandwidth would only shrink the gap. Therefore we expect the ground states to remain fully polarized, assuming the bandwidth is not too large.

Our method also helps us to study the nature of the excitations, leading us directly to an (as far as we know) undiscovered connection between a two-dimensional, strongly interacting, topologically non-trivial system of electrons and one-dimensional Ising spin physics bearing the same topology.

This paper is organized as follows: in Sec. I, we lay out an analytic argument for the ground state of the aforementioned (repulsive) interacting quantum Hall problems, $\mathbb{Z}_2$ and spin-$\frac{1}{2}$ QH; in Sec. II we construct pseudopotentials for both scenarios by placing the problem onto a cylinder, which makes one direction compact and reduces the 2D continuum problem to an effective 1D discrete chain; following this, in Sec. III, we use DMRG to solve for both the ground state manifolds of the constructed pseudopotentials and the system size scaling of the energy gap. In Sec. IV, we study the nature of the excited states for smaller cylinders using ED, and show an interesting property of this model – product states in the eigenspectrum, the number of which grows with system size – and use this property to discuss a connection between our model and the physics of 1D Ising spin chains. We choose to use naturalized units such that $e = \hbar = c = 1$, and set the mass of every particle to $m = 1$. The magnetic length is $l_B = B^{-1/2}$. For the purposes of DMRG and ED, we set $l_B = 1$.

## II. QUANTUM HALL POLARIZATION

When a uniform magnetic field $B$ is applied to a sample of electrons, the sample can be understood to be "discretized" in the sense that the non-interacting single-particle states, i.e Landau levels (not including spin or flavour) take on an integer degeneracy $N$ which is equal to the division of the sample area $A$ by the area of the quantized cyclotron orbit $2\pi l_B^2$. As the magnetic field is increased, the area of the quantized cyclotron orbits shrinks, both driving up the degeneracy of the Landau levels and increasing the energy gap $\omega$ between them [35, 37, 39]. Naturally, electronic interactions would intermix these single-particle states; however, if the magnetic field is large enough, the resulting gap between Landau levels may dominate over the electron-electron interactions, preventing Landau level mixing. Thus, in the presence of a strong enough magnetic field, the interacting many-body ground state of a partially or fully occupied lowest Landau level can be described without the need for higher Landau levels [35, 36, 39].

Given a large enough magnetic field, non-interacting electrons occupy a discrete set of LLL orbitals; therefore, a natural question to ask is: How does the filling, or number of electrons in the sample, affect the ground state in the presence of strong repulsive interactions? At $\nu = 1$, it is possible to place one particle per LLL orbital. Since the Landau-gauge orbitals are localized in at least one direction, and repulsive interactions minimize energy by driving particles apart, one might expect that at $\nu = 1$, the ground state would be just one particle per LLL orbital.

When electrons have no spin or additional flavour, the above scenario is fully filled and trivial; however, if the electrons have spin, it has been long understood that the repulsive interactions form a ferromagnet [39], often referred as the half-filled or $\nu = 1$ ferromagnet ($\nu = 2$ being fully filled) in literature [41].

By following similar variational arguments in Refs. [10, 17], we mathematically prove that a ground state of the spin-$\frac{1}{2}$ system at this filling is indeed one particle per site, and completely spin-polarized. The single-particle annihilation operator at position $\mathbf{r} = (x, y)$ within the sample area can be expanded in terms of all Landau level wavefunctions $\psi_a(x, y|m)$ – with increasing energy labelled by index $a$, spin $\beta \in \{\uparrow, \downarrow\}$, and Landau level orbital $m$ – as

$$c_\beta(\mathbf{r}) = \sum_{a=0}^{\infty} \sum_m \psi_a(x, y|m) d_{\beta, a, m}. \quad (1)$$

The Hamiltonian is $H = \hat{T} + \hat{V}$, where the kinetic energy $\hat{T}$ is diagonal in the Landau level single-particle basis Eq. (1), and can be decomposed into sum of contributions from each Landau level $\hat{T} = \hat{T}_0 + \hat{T}_1 + \cdots$. The difference between the lowest $\hat{T}_0$ and the first excited $\hat{T}_1$ Landau levels, $\omega$, is an important scale in this argument. The magnetic field is tuned such that the gap $\omega$ is much larger than the interaction strength. Following this requirement, we constrain our variational state $|\psi\rangle$ to the sector of Fock space describing many-body states formed out of single-particle LLL wavefunctions, killing all contributions from the kinetic energy $\hat{T}$ except $\hat{T}_0$. The lat-

ter being a constant which can be gauged away. We are now left with the task of minimizing the 2-body interaction operator $\hat{V}$, which is written

$$\hat{V} = \sum_{\mathbf{r}\,\mathbf{r}'} c_\alpha^\dagger(\mathbf{r}) c_\beta^\dagger(\mathbf{r}') V(\mathbf{r} - \mathbf{r}') c_\beta(\mathbf{r}') c_\alpha(\mathbf{r}). \qquad (2)$$

However, at a commensurate filling where it is possible to place the same number of particles per site, the density-density representation is more facilitating. One might simply anticommute the fermion annihilation and creation operators in order to place Eq. (2) into the density-density form minus a quadratic piece

$$\hat{V} = \sum_{\mathbf{r}\,\mathbf{r}'} c_\alpha^\dagger(\mathbf{r}) c_\alpha(\mathbf{r}) V(\mathbf{r} - \mathbf{r}') c_\beta^\dagger(\mathbf{r}') c_\beta(\mathbf{r}')$$
$$- V(0) \sum_{\mathbf{r}} c_\alpha^\dagger(\mathbf{r}) c_\alpha(\mathbf{r}). \qquad (3)$$

In thermodynamic limit, the quadratic piece is a global constant and can be neglected (see Supplement-B). Thus, the ground state can be determined by minimizing the interaction in the density-density form, which we henceforth call $H'$. We take $H'$ into the momentum basis and act it on our variational state:

$$H' \left| \psi \right\rangle = \sum_{\mathbf{q}} \hat{n}(\mathbf{q}) \tilde{V}(\mathbf{q}) \hat{n}(-\mathbf{q}) \left| \psi \right\rangle, \qquad (4)$$

where $\hat{n}(\mathbf{q}) = \sum_{\mathbf{r}} e^{i\mathbf{q}\cdot\mathbf{r}} \hat{n}(\mathbf{r})$ is the Fourier transform of the density operator. Writing this operator in the momentum basis has a two-fold purpose. First, since $\hat{n}(-\mathbf{q})^\dagger = \hat{n}(\mathbf{q})$ in the momentum basis, the above becomes

$$= \sum_{\mathbf{q}} |\hat{n}(\mathbf{q})|^2 \tilde{V}(\mathbf{q}) \left| \psi \right\rangle,$$

which is the sum of positive semi-definite functions of $\mathbf{q}$ (i.e for non-zero values of the hermitian operator $|\hat{n}(\mathbf{q})|^2$ and $\tilde{V}(\mathbf{q}) \geq \tilde{V}(0) \geq 0$). Thus every term with a non-zero value of $|\hat{n}(\mathbf{q})|^2$ increases the total interaction energy. Secondly, in the momentum basis the density operator is a sum over terms quadratic in the fermion operators: $n_\mathbf{q} = \sum_{\mathbf{k}} c_\alpha^\dagger(\mathbf{k}+\mathbf{q}) c_\alpha(\mathbf{k})$. When such an operator is acted onto a state which is one particle per orbital, and all the same flavour (i.e spin polarized up $\left|\uparrow\uparrow\cdots\uparrow\right\rangle$ or down $\left|\downarrow\downarrow\cdots\downarrow\right\rangle$), all but the $\mathbf{q}=0$ terms vanish. Thus, $\left|\psi\right\rangle$, chosen to be flavour polarized with one particle per orbital, minimizes the Hamiltonian; and the sum of positive semi-definite terms collapses to a single term:

$$= |\hat{n}(0)|^2 \tilde{V}(0) \left| \psi \right\rangle.$$

This expression tells us that the ground state is one which prohibits inter-orbital scattering. Remembering that a large gap $\omega$ places a prohibitive energy cost to occupying higher Landau levels than LLL, the true ground state $\left|\psi\right\rangle$ must be one which is one particle per LLL orbital and polarized. These two fully polarized states are not the

only states in the ground state manifold [41], which (for an $N$-particle system) includes $N-1$ additional states that are eigenstates of $S^2$ with eigenvalue $\frac{N}{2}(\frac{N}{2}+1)$. These are the states which are connected to fully polarized states by the total spin raising and lowering operators $S_\pm$. One might imagine that other eigenstates of $S^2$ might be solutions to the Hamiltonian Eq. (2). However, different irreducible representations of the $SU(2)$ symmetry (i.e different eigenstates of $S^2$ with eigenvalues less than $\frac{N}{2}(\frac{N}{2}+1)$) are not connected by any such operation. Therefore, there is no requirement for them to be ground states.

If we work with spinless fermions in a system which includes Landau levels with opposite magnetic fields $\pm B$ (i.e $\mathbb{Z}_2$), the language of the argument would still hold. Thus the ground state manifold woud contain two states: one fermion per LLL of type $+B$, which we write $\left|++\cdots+\right\rangle$, and its $\mathbb{Z}_2$ partner $\left|--\cdots-\right\rangle$. As will be discussed in the following section, unlike the $\nu=1$ spin-$\frac{1}{2}$ QH, the $\mathbb{Z}_2$ scenario has no additional states, owing to its reduced symmetry.

## III. PSEUDOPOTENTIALS: FROM 2D TO 1D

Due to the uniform nature of the magnetic field, the underlying spatial symmetries of the sample – translational and rotational – are preserved. Therefore degenerate single-particle states are necessarily labelled, up to a choice of gauge, by the eigenvalues of their generators. The choice of gauge is unphysical and arbitrary, thus choosing a gauge which preserves translational symmetry in one direction, the Landau gauge [37], gives rise to a spectrum composed of degenerate manifolds of single-particle states labelled by momentum in one direction (say $k_y$). If we impose periodic boundary conditions in the $y$-direction, by wrapping our sample of area $A$ into a cylinder of circumference $L_y$, the centers of localization become uniformly spaced with incremental values of $l_B^2 k_y = \frac{l_B^2}{L_y} 2\pi n$; where $n \in \mathbb{Z}$ for an odd number of sites $N$ and $n \in \mathbb{Z} + \frac{1}{2}$ for even $N$ [38]. With this choice of geometry, adiabatically threading a flux quantum through the loop of the cylinder moves the center of a Landau level wavefunction into its neighbor's [40, 50]. The direction $(\pm\hat{x})$ in which the wavefunctions moves depends on the sign of the magnetic field, meaning that Landau level centers which experience opposite magnetic fields move in opposite directions. We make this explicit in defining the LLL wavefunction [16], which we write (up to a normalization $\mathcal{N} = (L_y l_B \sqrt{\pi})^{-1/2}$):

$$\phi_{\xi,k_y}(\mathbf{r}) = \mathcal{N} \exp\left(iyk_y - \frac{1}{2l_B^2}(x - \xi l_B^2 k_y)^2\right), \quad (5)$$

where $\xi = \pm 1$ indicates the sign of the magnetic field, i.e the Chern number of the LLL. By restricting the electrons to occupy only the LLL's, we obtain a picture reminiscent of a 1D chain (Fig 2), except where the chain

"sites" label degenerate single-particle LLL orbitals of increasing momentum $k_y$. Along the $x$-axis, the "sites" are Gaussian localized with a width $l_B$, and the Gaussian centers are spaced by a distance $\Delta x = 2\pi l_B^2/L_y$, which depends on the ratio $\gamma = l_B/L_y$ [36]. The size of the 1D chain is fixed by the total orbital number $N$, which in turn fixes the length of the cylinder to be $L_x = N\Delta x$.

Practically, this projection is done by defining the LLL-projected electron operator

$$\hat{c}_\alpha(\mathbf{r}) = \sum_{\xi,n} \phi_{\xi,k_y}(\mathbf{r}) d_{\alpha,\xi,n}, \qquad (6)$$

which replaces $c_\alpha(\mathbf{r})$ in Eq. (2). The LLL operator $d_{\alpha,\xi,n}$ destroys a fermion with spin $\alpha \in \{\uparrow,\downarrow\}$, sign of magnetic field $\xi$, and momentum $k_y = \frac{2\pi n}{L_y}$.

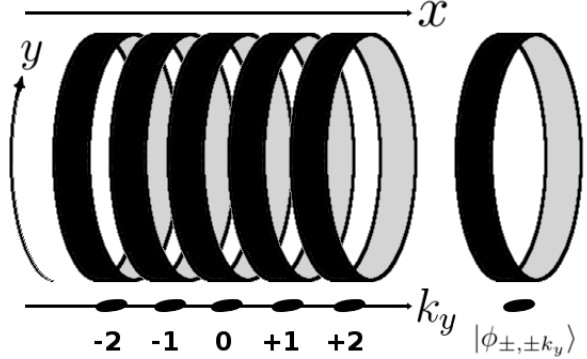

FIG. 2. The cylindrical sample of area $A = L_x L_y$ and circumference $L_y$, subdivided according to the LLL wavefunction centers which form the corresponding chain of LLL orbitals.

Naturally, the form of the interaction $\hat{V}$ upon projection depends on the problem: spin-$\frac{1}{2}$ QH or $\mathbb{Z}_2$. In the spin-$\frac{1}{2}$ case, all fermions feel the same sign of the magnetic field $\xi = +1$. Dropping the redundant label, the projected interaction is

$$H_{\frac{1}{2}\text{QH}} = \sum_{n,k,m} V_{km} d^\dagger_{\alpha,n+k} d^\dagger_{\beta,n+m} d_{\beta,n+m+k} d_{\alpha,n}, \quad (7)$$

where $n$ are integers (half-integers) if the number of sites is odd (even), and $k,m \in \mathbb{Z}$ always [38]. The matrix elements for a general projected two-body interaction are

$$V_{km} = \frac{\sqrt{\pi/2}}{L_y l_B} \int_{-\infty}^{\infty} d\tilde{x} \int_{-L_y/2}^{L_y/2} d\tilde{y} \qquad (8)$$
$$\times V(\tilde{x},\tilde{y}) e^{-i\tilde{y}\frac{2\pi k}{L_y}} e^{-\frac{1}{2l_B^2}(\tilde{x}+l_B^2\frac{2\pi m}{L_y})^2} e^{-\frac{1}{2}l_B^2(\frac{2\pi k}{L_y})^2}$$

with $(\tilde{x},\tilde{y}) \equiv (\mathbf{r}-\mathbf{r}')$.

We now turn to our toy model for TBG, where the opposite magnetic field LLL's are analog of the gauge-fixed hybrid Wannier states [16, 45, 46] in the chiral limit [15, 18]. For simplicity, we polarize the spin and make the system effectively spinless. We therefore drop the redundant spin index, and find the projected interaction to be

$$H_{\mathbb{Z}_2} = \sum_{n,k,m} V_{km} \Big( d^\dagger_{+,n+k} d^\dagger_{+,n+m} d_{+,n+m+k} d_{+,n}$$
$$+ d^\dagger_{-,-n} d^\dagger_{-,-n-m-k} d_{-,-n-m} d_{-,-n-k} \qquad (9)$$
$$+ 2d^\dagger_{+,n+k} d^\dagger_{-,-n-m-k} d_{-,-n-m} d_{+,n} \Big),$$

where $V_{km}$ is given in Eq. (8). By inspection, one can see that Eq. (9) conserves Chern number – a consequence of the sublattice polarization in the CSCM (Scenario 2 introduced in Introduction). Just as with the spin-$\frac{1}{2}$ QH Hamiltonian Eq. (7), only two flavours of fermions are present, except here the $SU(2)$ symmetry is reduced to the $\mathbb{Z}_2$ symmetry [29]. Following the argument laid out in Sec II, one ground state must be Chern polarized and one particle per LLL orbital: $|++\cdots+\rangle$. It is straightforward to check if the polarized state is an eigenstate of the projected Hamiltonian:

$$H_{\mathbb{Z}_2} |++\cdots+\rangle = \qquad (10)$$
$$\sum_{|n|\leq Q,|n+k|\leq Q} \Big( -V_{k,0} + V_{0,k} \Big) |++\cdots+\rangle,$$

noting that $Q \equiv (N-1)/2$ is the magnitude of the edge orbital momentum; but it is not so obvious that this is the lowest energy state. For that we implement DMRG and solve Eq. (9) numerically for the screened Coulomb interaction in the next section:

$$V_{sc}(\tilde{x},\tilde{y}) = \frac{g}{\sqrt{\tilde{x}^2+\tilde{y}^2}} \exp\Big( -\frac{\sqrt{\tilde{x}^2+\tilde{y}^2}}{l_s} \Big), \qquad (11)$$

where $g$ is the constant which sets the strength of the interaction. Since this constant multiplies the projected Hamiltonian (Eqs. (7) and (9)), we are free to rescale it without any consequence on the spectrum beyond a global rescaling of the energy.

However interestingly, there is a special case which can be shown exactly – short-range contact-like interactions, $V_\delta(\tilde{x},\tilde{y}) = g_\delta \, \delta(\tilde{x})\delta(\tilde{y})$. We say "contact-like" because it treats inter-sublattice scattering as scattering at a point, which can be understood as a highly-screened Coulomb interaction [?]. For this choice of interaction, the matrix element Eq. (8) becomes symmetric, i.e $V_{km} = V_{mk}$, and thus by Eq. (10), the energy of the polarized state is zero, as expected since like-flavour fermions are forbidden by Pauli exclusion from contact scattering. Conveniently, this choice of interaction is mathematically simpler to study than the screened Coulomb interaction, having an exact formula,

$$V_{km}^\delta = g_\delta \frac{\sqrt{\pi/2}}{L_y l_B} \exp\Big( -\frac{1}{2}(2\pi\gamma)^2 \big(m^2+k^2\big) \Big), \qquad (12)$$

and because the ground state of the repulsive interacting problem is clearly marked by zero energy.

## IV. DMRG RESULTS

Mixed real-momentum space representations of the cylinder geometry, where the momentum in $y$-direction is used as a conserved quantum number, while keeping the locality in the $x$-direction, have been shown to greatly reduce the required computational time and memory for employing DMRG [53]. The non-trivial topology of the LLL takes this a step farther, tying together the position and momentum via a single quantum number ($k_y$), and effectively collapsing the problem down to a 1D system [38, 50].

Such approaches are well motivated for QH by exact MPS representations discovered for QH systems – including the Laughlin, Moore-Read [55], and Haldane-Rezayi states [56] – and have been used to numerically calculate and characterize fractional QH states [47, 48, 50].

Unlike fractional QH ground states, the ground states of Eq. (7) and Eq. (9) at *integer* filling are much simpler. This is because they have a bond dimension equal to 1 for the fully polarized states, and it scales linearly with the system size for those states of Eq. (7) connected to the polarized states by symmetry.

While the dimensional reduction of the QH problem in this guiding-center representation is computationally less costly, it comes at the cost of longer range (Gaussian localized [50]) interactions in the effective 1D picture, even for short range interactions in real space, e.g contact-like $V_\delta(\tilde{x}, \tilde{y})$. The range of these Gaussian localized interactions grows with $\gamma^{-1}$, therefore, if the number of orbitals in the $x$-direction is not large enough, one risks modifying the form of the interaction by its truncation due to the system size [47]. This truncation error is a consequence of constructing a finite-size system using LLL's which extend to infinity (see Supplement B and C). As far as the error from this truncation is concerned, given there is one particle per every LLL orbital, we find that the truncation error from having too few LLL orbitals appears to have no effect on the ground state manifold. However, for sufficiently large enough screening length, we find that the one-particle-per-site order melts, clumping electrons at the edges and center of the cylinder. Nonetheless, the fully polarized states remain eigenstates, albeit at higher energy (see Supplement D for data on longer range interactions).

In order to better understand the system size scaling, and additionally to check for the presence of a gap, we use DMRG and determine the first excited states. We then plot the gap as a function of both increasing total number of orbitals and $\gamma^{-1}$. As shown for $l_s = 1l_B$ in Fig. 3a, we find there is a sufficiently large enough $N$ beyond which the gap saturates to a finite value, implying the presence of a gap in the $N \to \infty$ limit. The value at which the gap saturates to appears to be unchanged as we increase the cylinder circumference from $L_y = 8l_B$ to $20l_B$. This likewise suggests the gap saturates in the $L_y \to \infty$ limit, assuming $N$ is thermodynamically large.

The nature of the first excited states depends on $\gamma^{-1}$,

which controls the overlap of the Gaussian wavefunctions. For $1 < \gamma^{-1} < 2\pi$, the first excited states are product states of form

$$|\psi\rangle_Q = d^\dagger_{-,-Q} d_{+,Q} |+ + \cdots +\rangle, \qquad (13)$$

where $Q$ is the momentum index of the edge orbital. Likewise, there exists degenerate states related to it by the $\mathbb{Z}_2$ symmetry. Because the guiding centers of opposite-Chern LLL's wind in opposite directions, Eq. (13) contains no more than one particle per guiding center. However, once $\gamma^{-1}$ passes $2\pi$, the centers of neighboring Gaussian localized sites move within each other's standard deviation, hence the wavefunctions overlap significantly. This changes the site resolution, where two previously distinct sites can no longer be differentiated easily and act as if they are one site. Consequently, higher excited states containing two particles per guiding center, $d^\dagger_{-,-Q} d_{+,Q-1} |\cdots + +\rangle$ and $d^\dagger_{-,-Q+1} d_{+,Q} |\cdots + +\rangle$, are no longer necessarily energetically disfavored by the Coulomb repulsion against states with one particle per center, as the two guiding centers now substantially overlap. For $\gamma^{-1} > 2\pi$, a superposition of these states, $d^\dagger_{-,-Q} d_{+,Q-1} |\cdots + +\rangle + w \ d^\dagger_{-,-Q+1} d_{+,Q} |\cdots + +\rangle$, becomes energetically favorable over the product state Eq. (13). Not only does this relative weight $w$ depend on $\gamma^{-1}$, but for larger $\gamma^{-1}$ entanglement of the first excited states spreads over more sites at the edge. Fig. 4 compares the energies of a lowest-energy product state and a first excited state with respect to $\gamma^{-1}$ for small orbital numbers of $N = 6 - 8$ computed via ED. While for $\gamma^{-1} \lesssim 2\pi$ the first excited states are product states, increasing $\gamma^{-1}$ causes entangled states to be energetically favorable. Furthermore, we find that the manifold of first excited entangled states is characteristically similar to the manifold containing product state Eq. (13). Whether $\gamma^{-1}$ is above or below $\sim 2\pi$, the first excited states (and their $\mathbb{Z}_2$ partners) can be described as edge states with total Chern number two less than the fully polarized states, i.e $N - 2$.

For $H_{\frac{1}{2}\mathrm{QH}}$, DMRG converges to all $N + 1$ states in the ground state manifold, which includes both fully polarized states, and those connected by the symmetry $S_\pm$. As expected analytically, the ground state manifold is an irreducible representation of the $SU(2)$ symmetry, with eigenstates of $S^2$ and associated eigenvalues $\frac{N}{2}(\frac{N}{2} + 1)$. Just as with the ground states, the excited state manifold is an irreducible representation of the $SU(2)$ symmetry too, except with eigenvalues $\frac{N-2}{2}(\frac{N-2}{2} + 1)$. Unlike Eq. (13), the change in spin is not strictly localized to the edge, and the gap appears to shrink with increasing $\gamma^{-1}$ for both interactions, Figs. 5a-5b). Moreover, with increasing $N$ the excited states converge to one particle per site on average, which is also true for the ground states. This behavior alongside the vanishing gap holds for screening length $l_s = 1l_B$ – which has an extent of approximately 2 orbitals for $L_y = 10l_B$. We suspect that the ground state and excited state manifolds become degenerate in the $N \to \infty$ limit, further increasing the de-

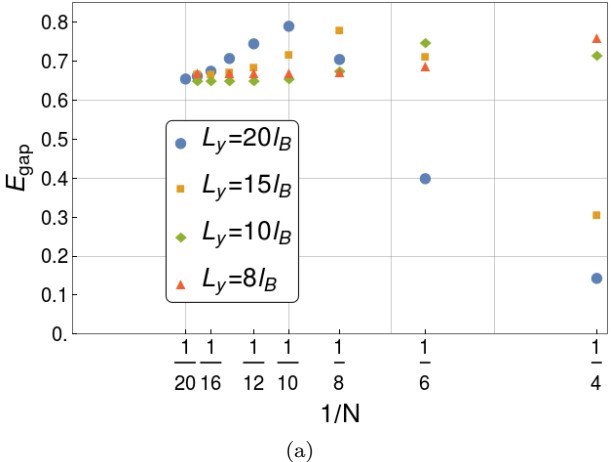
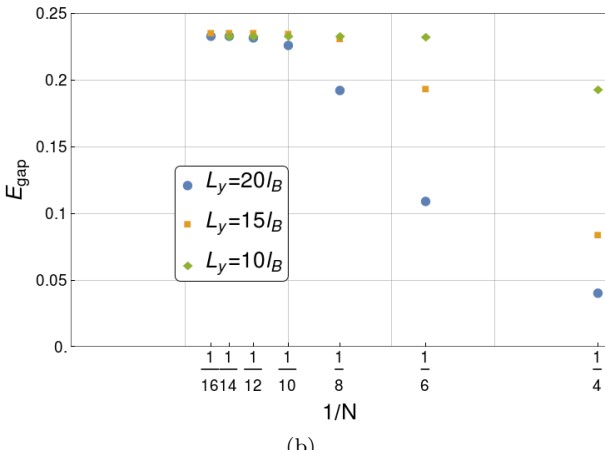

(a)

(b)

FIG. 3. The energy gap between the ground and first excited state manifolds, for various $\gamma^{-1} > 2\pi$ (see legends) and (a) with screened Coulomb interaction $l_s = 1$ (b) contactlike interaction in $\mathbb{Z}_2$ Hamiltonian. For sufficiently large enough $N$, the excited state is an edge state with total Chern $N - 2$. The energy converges to a value which is independent of $L_y$. For case (a), if $N$ is not large enough, the system is plagued by the truncation of finite size interaction terms, which become vanishingly small ($< 1\%$, see Fig 9 in Supplement C) alongside the appearance of the Chern $N - 2$ state. DMRG is performed with a cutoff of $10^{-8}$ and a maximum allowed bond dimension of 300, yet neither the ground states nor first excited states reach this limit. We set the upper limit on the Hamiltonian MPO bond dimension to be 5000 with a $10^{-13}$ cutoff.

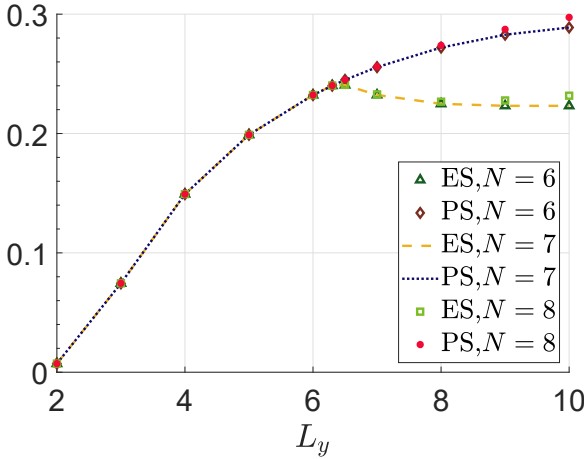

FIG. 4. The comparison between the energies of a lowest-lying product state (PS) and a first excited state (ES) with respect to $\gamma^{-1} = L_y/l_B$ for small total orbital numbers ranging between $N = 6 - 8$ computed via exact diagonalization. The plot shows that the first excited states are product states for $\gamma^{-1} \lesssim 2\pi$, whereas they are entangled for $\gamma^{-1} \gtrsim 2\pi$.

generacy of the ground state which is already extensive in the system size.

## V. EXCITED STATES AND EMERGENT SPIN PHYSICS

Despite the numerical advantage afforded by the Landau gauge, the resulting projected Hamiltonian is by no means trivial. For starters, the kinetic energy is implicit in the construction of the model from the projection onto the LLL; and likewise, the Hamiltonian Eq. (9) is no simple sum of local density operators, instead bearing a non-trivial structure with correlated "hopping" across the chain of orbitals, which is deeply related to the momentum conservation around the cylinder. As such, one might not expect the ground state to be a product state, yet as we discussed in Sec II and shown numerically, product states are a natural consequence of strong interactions in these systems at integer filling. Given that these are truly two-dimensional systems, one would expect the entanglement in the ground state to follow area law, and scale linearly with the system size; and this is indeed the case for the states (except the polarized states) in the ground state manifold of the $SU(2)$ QH ferromagnet. However, when the symmetry is reduced to $\mathbb{Z}_2$, the only states in the ground state are the polarized states, and thus area law appears to be beaten. Likewise, there exists excited states, such as the edge state Eq. (13), which might exhibit better than area law scaling.

It is then natural to ask if product states exist throughout the spectrum, what their fraction to the dimension of the Hilbert space is, and if the number of those states increases with the system size $N$ and ratio $\gamma^{-1}$. In order to answer these questions, we constrain ourselves to contact interactions on smaller cylinders (with $l_B = 1$) computed by ED. We find that the excited states do exist throughout the spectrum, but that their fractions depend significantly on the guiding center overlaps, which is controlled by $\gamma^{-1}$. More precisely, by plotting the inverse participation ratio, $P_\alpha = \sum_n |\psi_{\alpha n}|^4$ for eigenstate $\psi_\alpha$, we observe that the fraction of total number of prod-

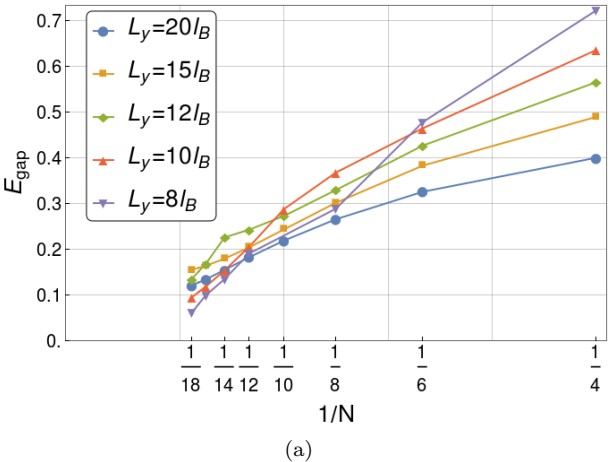

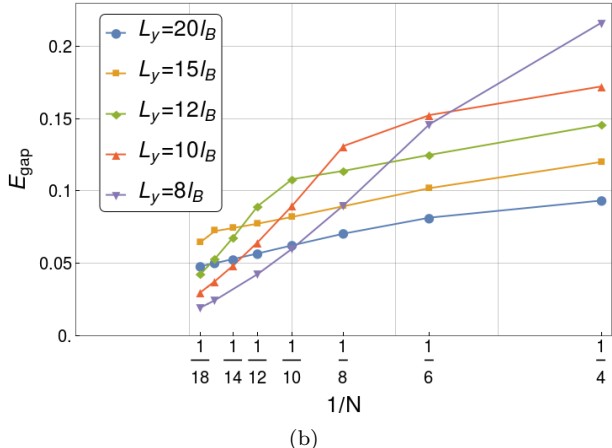

$$\text{(a)} \qquad\qquad\qquad\qquad \text{(b)}$$

FIG. 5. The energy gap between the ground and first excited state manifolds, for various $\gamma^{-1} > 2\pi$ (see legends) and (a) with screened Coulomb interaction $l_s = 1$ (b) contactlike interaction for $H_{\frac{1}{2}\mathrm{QH}}$ Hamiltonian. For both cases, the gap shrinks as the total number of orbitals $N$ increases, suggesting a larger degeneracy in the ground state manifold in the thermodynamic limit. DMRG is performed with a cutoff of $10^{-8}$ and a maximum allowed bond dimension of 1000; neither the ground states nor first excited states surpass these limits on bond dimension. The limits on the MPO bond dimension are the same as for Fig 3a-3b.

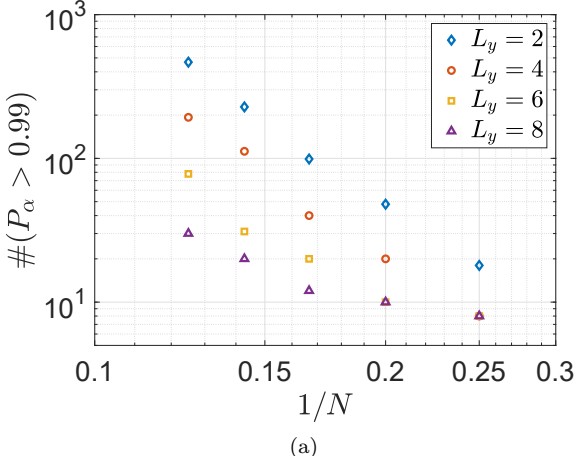

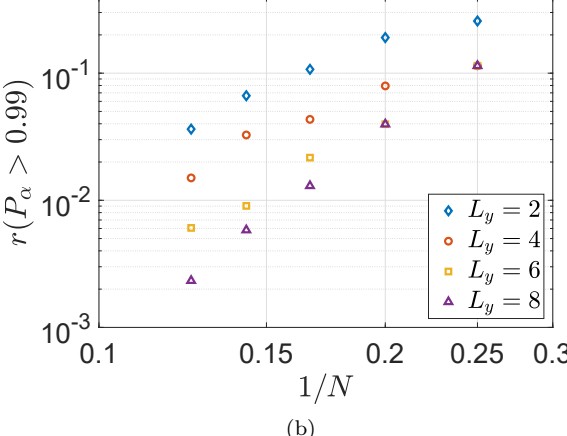

$$\text{(a)} \qquad\qquad\qquad\qquad \text{(b)}$$

FIG. 6. (a) The number of states with inverse participation ratio $P_\alpha > 0.99$ as a function of total orbital number $N$. (b) The ratio of the number of states with inverse participation ratio $P_\alpha > 0.99$ to the dimension of the Hilbert space, as a function of $N$. In both subfigures, we use contactlike interaction for $Z_2$ Hamiltonian. Although the number of product states increase with the total orbital number $N$ as expected, the ratio of it to the dimension of the Hilbert space shrinks. Additionally increasing the circumference $\gamma^{-1}$ decreases the number of product states as well as the ratio. Thus in the thermodynamic limit, the presence of product states is negligible.

uct eigenstates to Hilbert space dimension shrinks with increasing $\gamma^{-1}$ (Fig 6b), and yet the number of product states does not vanish (Fig 6a). Additionally, we observe that for $\gamma^{-1} \lesssim 2\pi$, the number of product states increases with a greater slope as a function of $1/N$ compared to $\gamma^{-1} \gtrsim 2\pi$. On the other hand, the decrease in the ratio $r(P_\alpha > 0.99)$ as a function of $1/N$ is faster for $\gamma^{-1} \gtrsim 2\pi$ with almost two decades of magnitude traversed compared to $\gamma^{-1} \lesssim 2\pi$ with only one decade for the same range of $N$.

For this to become clear, let us consider the limit $\gamma^{-1} \ll 2\pi$, where the LLL guiding centers only marginally overlap with their neighbors. From ED, we know that the spectrum is composed entirely out of product states in this limit. Here the exact nature of the excited states found via ED suggests an interesting connection to 1D Ising-type spin physics, with Chern number playing the role of spin. As a prerequisite for this connection, we remember that a Chern $C = \pm 1$ Landau orbital at $\pm n$ are located together in real space, both having Gaussian centers at $x = 2\pi\gamma l_B n$ (see Fig 7). This is because the real-space guiding centers of the

opposite-Chern LLL's wind in opposite directions about the cylinder. Thus, we consider each LLL orbital with $C = \pm 1$ at momentum $\pm n$ to act as a *local* spin (in real space), mapped to local spin states as: $|+\rangle_{+n} \longrightarrow |\uparrow\rangle_n$ and $|-\rangle_{-n} \longrightarrow |\downarrow\rangle_n$. Each is an eigenstate of the local Pauli-Z Chern-pseudospin operator

$$Z_n = \frac{1}{2}\left(d^\dagger_{+,n} d_{+,n} - d^\dagger_{-,-n} d_{-,-n}\right) \tag{14}$$

with eigenvalue $\pm 1/2$. For example, Eq. (13) for $N = 4$,

$$d^\dagger_{-,2} d_{+,-2} |{+}{+}{+}{+}\rangle = |0, +, +, \pm\rangle,$$

in LLL basis, corresponds to the many-body spin state $|\downarrow, \uparrow, \uparrow, \uparrow\rangle$ in real space. Understandably, since the electrons are not fixed spins, and double occupancy of orbitals with the same real-space guiding centers can occur, there exists states in the Fock space, such as $|-, +, +, +\rangle$, where $Z_{\pm 2} = 0$ and therefore do not have a corresponding local-spin state. Since states which do not follow this scheme necessarily have electrons at the same guiding center, they are guaranteed, because of the repulsive interactions, to be at a higher energy than those with non-zero eigenvalues of $Z_n$.

This is the scenario discussed in the previous section regarding the lowest excited state. In the limit $\gamma^{-1} \ll 2\pi$, neighboring edge orbitals at $Q$ and $Q - 1$ do not overlap in real space, and consequently a product state analogous to an Ising spin flip Eq. (13) is energetically preferred over those states with more than one particle per guiding center, e.g. $d^\dagger_{-,-Q} d_{+,Q-1} |{+}\cdots{+}{+}\rangle$ and $d^\dagger_{-,-Q+1} d_{+,Q} |{+}\cdots{+}{+}\rangle$. It is only once $\gamma^{-1} \sim 2\pi$ that the neighboring orbitals at $Q$ and $Q - 1$ overlap sufficiently enough such that these states can form a superposition which has relatively lower energy than Eq. (13).

This naturally describes why entangled states make up an increasingly larger fraction of the Hilbert space as we increase $\gamma^{-1}$. Because edge orbitals with more than one particle per real space center can lower their energy through entanglement with their neighbors, given their wavefunctions sufficiently overlap. This being said, those product states which individually fall within the Ising scheme, such as Eq. (13), remain eigenstates. Thus, in the language of the Ising model, the number of product eigenstates increases with the number of possible "spin flips", which increases with the total number of LLL orbitals $N$, as shown in Fig 6a.

This apparent correspondence between eigenstates and local spin moments motivates a different way of thinking about the model as a whole, where intuition about 1D models of local moments, such as the Ising model, translates into something physically meaningful in the 2D $\mathbb{Z}_2$ QH problem. In the Supplement H, we detail how the Ising transverse field $\sum_i h X_i$, which pushes the spins into the xy-plane, corresponds (to leading order in $\gamma$) to a wall defect along the length of the cylinder in real space. Since opposite-Chern LLL with the same real-space center move in opposite directions about the cylinder, the largest contribution to colliding with the wall is to flip the Chern number at that guiding center, thus mixing the "spin". We do not explore the consequences of such a term to ground state physics, nor explore this correspondence beyond what we have discussed here, as this moves beyond the scope of this paper, and instead leave this for future research.

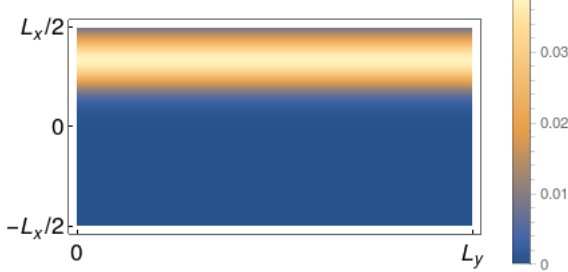

FIG. 7. Density of the LLL's (Eq. (5)) with $\xi = \pm 1$ in orbital $k_y = \pm 6\frac{2\pi}{L_y}$. Opposite Chern LLL share the same guiding-center as long as they have opposite momentum. Cylinder circumference is $L_y/l_B = 15$, with $N = 18$ LLL orbitals.

## VI. DISCUSSIONS AND CONCLUSIONS

The recent advances in stacked 2D materials, including TBG, have opened an entirely new avenue for research into the interplay of topology and strong interactions, previously only possible with metals in high magnetic fields. In this paper, we provide a simple model, constructed out of continuum lowest Landau levels, which captures the essential physics of two strongly interacting flat bands when the bands have identical ($H_{\frac{1}{2}QH}$) or opposite ($H_{\mathbb{Z}_2}$) Chern number, and we study it at integer filling. We find analytically (Sec II) for either problem that the ground state is a fully-polarized state (or symmetry related); which agrees with numerics on a finite cylinder given the interaction range is not too large. For the $\mathbb{Z}_2$ case, we find that the gap at $l_s = 1l_B$ saturates to a value independent of the system size, but shrinks with increasing range of the interactions. For large enough $l_s$, the gap collapses, and the ground state one-particle-per-site order melts into a state with greatest density at the center and edges. We find, however, that the polarized states remain eigenstates even when the interactions are longer ranged, all of which suggests the breakdown of order may be a consequence of the open boundary conditions. If instead the spin-$\frac{1}{2}$ QH case is considered, the gap appears to shrink with increasing $N$, and the first excited states compose an irreducible representations of the $SU(2)$ symmetry, which converges to one-particle-per-site order with increasing $N$. We expect this state to merge with the ground state manifold in the $N \longrightarrow \infty$ limit.

Let us take the opposite spins of the spin-$\frac{1}{2}$ QH problem to represent opposite layers of the bilayer QH problem. If we introduce an inter-layer interaction anisotropy which bias one layer over another, the $SU(2)$ symmetry breaks, splitting the extensive degeneracy of the ground state manifold and selecting out a spin(layer)-coherent ground state (see Supplement G). Such an instability toward a coherent state does not exist for repulsion between bands of opposite Chern number, where the ground state is only two-fold degenerate between spontaneously $\mathbb{Z}_2$-symmetry broken states [29], and numerically appears to be separated by a gap from the rest of the spectrum. These results are in agreement with mean field theory [16] and provide a simple explanation for the stability of a Chern-polarized ground state.

The primary purpose of our Manuscript is to provide a simple model for the study of strongly interacting problems in electronic systems with non-trivial topological bands and net-zero Chern number. More broadly, this work is meant to inspire further study [18] of crystals with topologically non-trivial Chern bands. In these scenarios, one can take advantage of the fact that it is always possible to construct maximally localized functions in 1D, and by extension, a gauge can be chosen in 2D such that single-particle wavefunctions are maximally localized along one direction [16, 45]. As with the LLL wavefunction, the real-space centers of these hybrid Wannier states are adiabatically connected through varying momenta, and thus there exists a dimensional reduction from 2D to 1D advantageous to numerics. Original constructions for such single-particle states [45] were suitable only in the continuum limit, and suffered from a gauge freedom which made the practical description of many-body states, such as the lattice analog to the FQH, difficult – plagued by insignificant overlaps between exact FQH ground states on a lattice and those constructed from the Laughlin wavefunction using hybrid Wannier states [46]. This was later remedied by *Wu, Regnault, and Bernevig* [46] through a procedure which trades off exact 1D maximal localization for exponential localization, but produces an orthonormal basis which can be gauge-fixed such that they behave like Landau orbitals. Very recently, such gauge-fixed hybrid Wannier states for Chern bands have been used to study strong interactions in TBG [18]. In general, this approach allows a more natural study of the MHS, capturing those details, such as a finite bandwidth and detailed shape of the Wannier functions, not available to the LLL.

The success of the DMRG technique applied to the models studied here is in part due to the low bond dimension of the ground states, which was true independent of our choice of interaction. This is mainly because of the preference of the repulsive interactions to choose a state which suppresses inter-orbital scattering, when the filling is such that one particle can be placed per single-particle orbital (i.e an integer filling). Such low bond-dimension ground and excited states might arise in more generic strongly repulsive systems of this kind at integer filling, which would make these numerical approaches an invaluable tool in the exploration of the plethora of new devices hosting topologically non-trivial narrow bands, as well as studying interacting QH states, such as the $\nu = 1$ QH ferromagnet [39, 41] studied here. This could include extending our study of the $\nu = 1$ QH state by including bilayer-type anisotropies or Zeeman terms not included in Eq. (7), which would provide a fertile platform for studying exciton condensation in bilayer devices [41, 42] or probing the lowest energy excitations of the $\nu = 1$ QH [39].

Furthermore, our $\mathbb{Z}_2$ Hamiltonian may open up a pathway toward testing recently proposed connections between $\mathbb{Z}_2$ topological order, thermalization [57], and information scrambling [58] in two-dimensions. As shown in a previous work [58] authored by one of us, the degeneracy structure of a spectrum has profound effects in dynamics of information scrambling probed by out-of-time-order correlators (OTOC). Because the ferromagnetic regime of the 1D transverse-field Ising model (TFIM) is dual to a 1D topological superconductor (the Kitaev chain), the presence of edge states results in a topological degeneracy which exists throughout the spectrum, and such a degeneracy inhibits information scrambling of edge operators, even at infinite temperature. How this extends to higher than one dimension has yet to be studied, in part hindered by the growth of the bond dimension in time. We've shown in our manuscript that our integer-filled $\mathbb{Z}_2$ Hamiltonian on open cylinders likewise exhibits topological degeneracy throughout the spectrum; and because of its connection to 1D Ising spin physics, as well as the presence of low-bond dimension excited states, our model may be one of the best candidates for numerically exploring information scrambling and in general nonequilibrium quantum physics in two dimensions.

As a final point, if one were to introduce a single-particle potential to Eq. (9) which biases one Chern-flavour over the other, then in the limit where one flavour electron is prohibitively expensive to create, Eq. (9) reduces to the pseudopotentials for the well known spin-polarized QH [53]. Thus at fractional filling $\nu = 1/3$ there exists the famous FQH state well-described by the Laughlin wavefunction [20, 40]. Seeing that no zero-field FQH state has yet been observed in TBG, it is an interesting question to ask what is the fate of the FQH at $\nu = 1/3$ when the bias is taken to be zero, i.e approaching $\mathbb{Z}_2$.

## VII. ACKNOWLEDGEMENTS

We would like to thank Jian Kang and Kai Sun for helpful discussion, and we are especially thankful to Oskar Vafek for his invaluable insight and many helpful discussions. P. Myles Eugenio is supported by NSF DMR-1916958. Ceren B. Dağ is supported by NSF Grant EFRI-1741618. The DMRG calculations performed here used the Intelligent Tensor C++ library (version 2.0.11)

[59].

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

## VIII.   SUPPLEMENTARY

### A.   Derivation of $H_{\mathbb{Z}_2}$

As discussed in the Introduction, there are two scenarios which inspire the study of TBG using LLL wavefunctions. Just as with the main text, we focus on, and use the language of the valley-polarized Scenario (2) in constructing the toy model. The valley-polarized Scenario (2) can be understood in analogy with the continuum model of TBG [8], where opposite-sign magnetic field LLL's play the role of opposite-sublattice hWS's (hybrid Wannier states) in the chiral limit [18]. Thus, we construct this model using continuum LLL's [37]. Working in the Landau gauge [37], the LLL wave functions are

$$\phi_{\xi,k}(\mathbf{r}) = \mathcal{N}e^{iky}e^{-\frac{1}{2l_B^2}(x-\xi l_B^2 k)^2}, \tag{15}$$

which are localized along the $x$-axis and compact about the cylinder's circumference ($L_y$) in $y$-direction. The (spinless/spin-polarized) projected single-particle annihilation operator is thus

$$\hat{c}(\mathbf{r}) = \sum_{k,\xi} \phi_{\xi,k}(\mathbf{r})d_{\xi,k}, \tag{16}$$

where $d_{\xi,k}$ annihilates a fermion with sign of magnetic field $\xi = \pm 1$ at LLL orbital $k$, as described in Sec III. Opposite magnetic field LLL are defined to be on opposite sublattices at any given point $\mathbf{r}$, and orthogonal by construction. This leads us to consider only operators which necessarily conserve Chern number $\xi$, i.e

$$\sum_{\mathbf{r}} \hat{c}^\dagger(\mathbf{r})f(\mathbf{r})\hat{c}(\mathbf{r}) = \sum_{\xi}\sum_{kp} d^\dagger_{\xi,k}d_{\xi,p}\left(\sum_{\mathbf{r}}\phi^*_{n,k}(\mathbf{r})\phi_{n,p}(\mathbf{r})f(\mathbf{r})\right). \tag{17}$$

Likewise, if instead we considered Scenario (1), the inverse Moire length scale $l_M^{-1}$ is significantly smaller than the momentum-space separation between electronic states in opposite valleys $a^{-1}$ (where $a$ is the graphene lattice constant). Thus, for the long-range Coulomb interaction, scattering between valleys is highly suppressed relative to scattering within a valley [10, 16, 30], and as with Scenario (2), we need keep only Chern number conserving terms in the interaction. Whatever the case, we construct the pseudopotentials by projecting the interaction operator onto these chosen single-particle states:

$$\sum_{\mathbf{r}\,\mathbf{r}'} : \hat{c}^\dagger(\mathbf{r})\hat{c}(\mathbf{r})V(\mathbf{r}-\mathbf{r}')\hat{c}^\dagger(\mathbf{r}')\hat{c}(\mathbf{r}') :$$

$$= \sum_{mn}\sum_{k_y p_y k'_y p'_y} : d^\dagger_{m,k_y}d_{m,p_y}d^\dagger_{n,k'_y}d_{n,p'_y} : \left(\sum_{\mathbf{r}\,\mathbf{r}'}\phi^*_{m,k_y}(\mathbf{r})\phi_{m,p_y}(\mathbf{r})V(\mathbf{r}-\mathbf{r}')\phi^*_{n,k'_y}(\mathbf{r}')\phi_{n,p'_y}(\mathbf{r}')\right)$$

$$= \sum_{k_y p_y k'_y p'_y} : d^\dagger_{+,k_y}d_{+,p_y}d^\dagger_{+,k'_y}d_{+,p'_y} : \left(\sum_{\mathbf{r}\,\mathbf{r}'}\phi^*_{+,k_y}(\mathbf{r})\phi_{+,p_y}(\mathbf{r})V(\mathbf{r}-\mathbf{r}')\phi^*_{+,k'_y}(\mathbf{r}')\phi_{+,p'_y}(\mathbf{r}')\right)$$

$$+ \sum_{k_y p_y k'_y p'_y} : d^\dagger_{-,k_y}d_{-,p_y}d^\dagger_{-,k'_y}d_{-,p'_y} : \left(\sum_{\mathbf{r}\,\mathbf{r}'}\phi^*_{-,k_y}(\mathbf{r})\phi_{-,p_y}(\mathbf{r})V(\mathbf{r}-\mathbf{r}')\phi^*_{-,k'_y}(\mathbf{r}')\phi_{-,p'_y}(\mathbf{r}')\right)$$

$$+ 2\sum_{k_y p_y k'_y p'_y} : d^\dagger_{+,k_y}d_{+,p_y}d^\dagger_{-,k'_y}d_{-,p'_y} : \left(\sum_{\mathbf{r}\,\mathbf{r}'}\phi^*_{+,k_y}(\mathbf{r})\phi_{+,p_y}(\mathbf{r})V(\mathbf{r}-\mathbf{r}')\phi^*_{-,k'_y}(\mathbf{r}')\phi_{-,p'_y}(\mathbf{r}')\right), \tag{18}$$

taking note of the normal ordering. It is not necessary to calculate all three matrix elements, as they are related to the first by symmetry. We use the fact that $\phi_{-,k}(\mathbf{r}) = \phi^*_{+,-k}(\mathbf{r})$, and find

$$
= \sum_{k_y p_y k'_y p'_y} : \left( d^\dagger_{+,k_y} d_{+,p_y} d^\dagger_{+,k'_y} d_{+,p'_y} + d^\dagger_{-,-p_y} d_{-,-k_y} d^\dagger_{-,-p'_y} d_{-,-k'_y} + 2 d^\dagger_{+,k_y} d_{+,p_y} d^\dagger_{-,-p'_y} d_{-,-k'_y} \right) :
$$

$$
\times \sum_{\mathbf{r}\mathbf{r}'} \left( \phi^*_{+,k_y}(\mathbf{r}) \phi_{+,p_y}(\mathbf{r}) V(\mathbf{r}-\mathbf{r}') \phi^*_{+,k'_y}(\mathbf{r}') \phi_{+,p'_y}(\mathbf{r}') \right)
$$

Writing the sums over $\mathbf{r}$ and $\mathbf{r}'$ as an integral over center-of-mass $(X,Y)$ and relative coordinates $(\tilde{x},\tilde{y})$, we find that the explicit form of the matrix element takes the form

$$
\int d\mathbf{r}\, d\mathbf{r}'\, \phi^*_{+,k_y}(\mathbf{r}) \phi_{+,p_y}(\mathbf{r}) V(\mathbf{r}-\mathbf{r}') \phi^*_{+,k'_y}(\mathbf{r}') \phi_{+,p'_y}(\mathbf{r}')
$$

$$
= \mathcal{N}^4 \int dY\, e^{iY(-k_y+p_y-k'_y+p'_y)} \int dX d\tilde{y} d\tilde{x}\, e^{i\frac{\tilde{y}}{2}(-k_y+p_y+k'_y-p'_y)} V(\tilde{x},\tilde{y})
$$

$$
\times \exp\left( -\frac{2}{l^2}(X - \frac{l^2}{4}(k_y+p_y+k'_y+p'_y))^2 \right)
$$

$$
\times \exp\left( -\frac{1}{2l^2}(\tilde{x} + \frac{l^2}{2}(-k_y-p_y+k'_y+p'_y))^2 \right)
$$

$$
\times \exp\left( -\frac{l^2}{8}(k_y-p_y+k'_y-p'_y)^2 \right) \times \exp\left( -\frac{l^2}{8}(k_y-p_y-k'_y+p'_y)^2 \right),
$$
(19)

which explicitly depends only on two momenta, as the others vanish upon integrating over the center-of-mass coordinates. If we then choose the following coordinates:

$$
k_y = \frac{2\pi}{L_y}(n+k),
$$

$$
p_y = \frac{2\pi}{L_y}n,
$$

$$
k'_y = \frac{2\pi}{L_y}(n+m),
$$

$$
p'_y = \frac{2\pi}{L_y}(n+m+k),
$$

we obtain Eq. (8), up to a constant multiplier. It is important to note that the operators in Eq. (7) and Eq. (9) conserve momentum, which appears explicitly in Eq. (19) as a delta-function upon integrating over $Y$.

As a final point, note that opposite-sign magnetic field fermions are Kramer partners under $C_2T$, and that consequently the LLL guiding centers of the effective 1D chain (Fig 2) are inverted about the origin relative to its Kramer partner, such that the operators $d^\dagger_{+,k_y}$ and $d^\dagger_{-,-k_y}$ both create an electron at $x = l^2_B k_y$ but with opposite momenta.

## B. The LLL-projected number operator $\hat{N}$, Normalization, and Confining Potentials

In Section II we discussed a variational argument for the ground states of the $\nu = 1$ spin-$\frac{1}{2}$ QH and $\mathbb{Z}_2$. This argument is precedent upon the fact that the total number operator

$$
\hat{N} = \sum_{\mathbf{r}} c^\dagger(\mathbf{r}) c(\mathbf{r})
$$
(20)

is proportional to the identity upon projection to an $N$-particle state of LLL's. If we consider a finite cylinder of length $L_x$, and use Eq. (5) with $\xi = +1$ and normalization $\mathcal{N}^{-1} = \sqrt{L_y l_B \pi^{1/2}}$, we find that the number operator is

$$
\hat{N} = \mathcal{N}^2 L_y \sum_n d^\dagger_{\alpha,n} d_{\alpha,n} \left( \int_{-\frac{L_x}{2}}^{\frac{L_x}{2}} dx\, e^{-\frac{1}{l^2_B}(x-2\pi l^2_B/L_y n)^2} \right).
$$
(21)

Where we see that when $L_x \to \infty$ holds,

$$
= \sum_n d^\dagger_{\alpha,n} d_{\alpha,n} = N\mathbb{1},
$$

and thus, the number operator behaves as a number operator in the thermodynamic limit. However otherwise, for finite $L_x$, the number operator does not properly count $N$ particles, instead behaving as a one-body potential. The potential depends explicitly on the momentum index $n$, shrinking for those LLL orbitals that are closer to the edge, and rapidly vanishing for orbitals outside the allowed region on the cylinder.

This problem is related to the fact that, even though the region of interest lies within the domain $x \in (-L_x/2, L_x/2)$, the LLL wavefunctions are technically solutions to an infinite system size problem, and therefore extend to infinity. In other words, the wavefunctions Eq. (5) are not the proper solutions for the single particle states constrained to the finite region. However, it is well understood that the degeneracy of the LLL is fixed by the area of the sample [39], and thus the length of the cylinder would constrain the number of LLL orbitals to be finite. This contradiction with the fact that the wavefunctions themselves extend to infinity is irrelevant if $L_x$ is (technically finite but) thermodynamically large, which in turn means $N$ is thermodynamically large; however, for small $N$, this argument does not hold, and one needs to consider a confining potential [48].

With this in mind, we ask if a confining potential exists such that when combined with the quadratic term of Eq. (3), the net effect is a constant which can be gauged away. Indeed a potential $V(0)U(x)$ – which is $U(x) = 1$ for $|x| > L_x/2$, and $U(x) = 0$ otherwise – does the trick:

$$
\begin{aligned}
V(0)\big(\hat{N} + \hat{U}\big) &= V(0)\mathcal{N}^2 L_y \sum_n d_{\alpha,n}^\dagger d_{\alpha,n} \Big( \int_{-\frac{L_x}{2}}^{\frac{L_x}{2}} dx \, e^{-\frac{1}{l_B^2}(x - 2\pi l_B^2/L_y n)^2} \\
&\quad + \int_{-\infty}^{-\frac{L_x}{2}} dx \, e^{-\frac{1}{l_B^2}(x - 2\pi l_B^2/L_y n)^2} + \int_{+\frac{L_x}{2}}^{+\infty} dx \, e^{-\frac{1}{l_B^2}(x - 2\pi l_B^2/L_y n)^2} \Big) \\
&= V(0)\mathcal{N}^2 L_y \sum_n d_{\alpha,n}^\dagger d_{\alpha,n} \int_{-\infty}^{\infty} dx \, e^{-\frac{1}{l_B^2}(x - 2\pi l_B^2/L_y n)^2} = V(0)N\mathbb{1}.
\end{aligned}
\tag{22}
$$

Since $V(0)$ is singular for Coulomb interaction, this would amount to a precisely chosen perfectly confining well.

Similarly, one can achieve the same goal by instead normalizing the LLL wavefunctions differently for different $n$, as

$$
\mathcal{N}_n^{-2} = L_y \int_{-\frac{L_x}{2}}^{\frac{L_x}{2}} dx \, e^{-\frac{1}{l_B^2}(x - 2\pi l_B^2/L_y n)^2}.
\tag{23}
$$

This does not change the shape of the wavefunctions, except in rescaling their amplitude such that the segment of the wavefunctions which lay in the region $x \in (-L_x/2, L_x/2)$ is normalized to unity. The effect of which is to increase the amplitude of those orbitals whose wavefunctions fall outside the desired region (see Fig. 8), which in turn forces the number operator to be proportional to unity. Physically, since $\mathcal{N}_n$ appears in the calculation of the interaction matrix elements (Eq. (19)), this has the effect of making scattering into orbitals outside the desired region prohibitively expensive.

In this paper, we do not normalize our wavefunctions using Eq. (23), instead choosing to follow previous literature [47–50], using a constant normalization across all orbitals. However, in excluding orbitals beyond our cylinder region, we are effectively introducing a potential which makes them energetically expensive to occupy, such that scattering into those orbitals can be neglected.

### C. Truncation Error

One might think that if all Landau levels were included, the projection of the interaction $V(\mathbf{r} - \mathbf{r}')$ would be unitary (and thus reversible). However, if the Landau levels extend to infinity, truncating the system size to finite $N$ ruins the reversibility, resulting in a "blurring" of $V(\mathbf{r} - \mathbf{r}')$.

As we discussed in Supplement-B, a confining potential can be included which makes orbitals outside those $N$ orbitals prohibitively expensive to occupy. This justifies the truncation of the orbital chain, as scattering processes into those LLL which lay outside the well are pushed up higher in energy.

Likewise, such a "blurring" due to truncation may also arise for infinite-length cylinders, where $N \longrightarrow \infty$. This "blurring" is unphysical, and introduced as a practical matter of regulating the range of the projected interaction, which would otherwise extend to infinity [50]. Nonetheless, since the LLL's are Gaussian localized, the strength of the interaction falls off like a Gaussian, and thus a sufficiently large cutoff can be chosen such that the effect of the truncation is marginal.

One would then expect that in our scheme – where the truncation is set by the number of LLL orbitals – a sufficient number of orbitals (for a given $\gamma^{-1}$) can be chosen such that the truncation error is small. In fact, the size of the

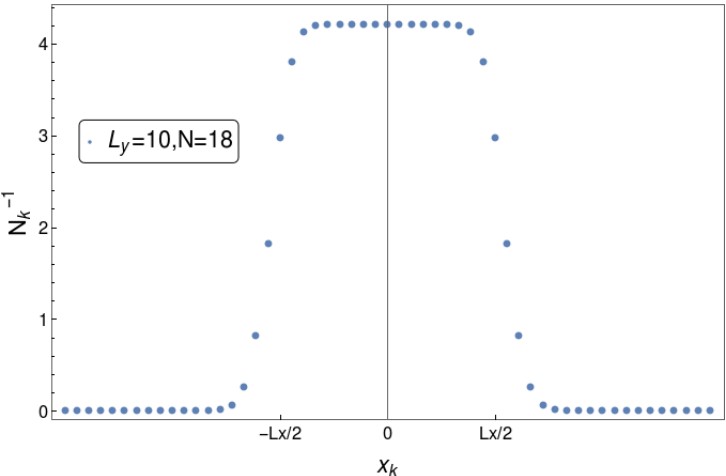

FIG. 8. The suggested alternative inverse momentum-dependent normalization $\mathcal{N}_n^{-1}$ as a function of the mean of the LLL orbital $x_n = 2\pi\gamma l_B n$.

interaction elements which are dropped shrinks considerably with increasing $N$ (Fig. 9), and is expected to vanish in the limit $N \to \infty$. By studying how physical observables scale with $N$, we determine the nature of the ground and excited states independent of the truncation error. Hence we find that there exists an $N$ beyond which physical observables associated with the first excited states begin to saturate (energy gap, density), and the value of the total Chern number is fixed. Here we provide an additional evidence by plotting the upper bound on the size of dropped interaction elements as a function of $N$, Fig. 9.

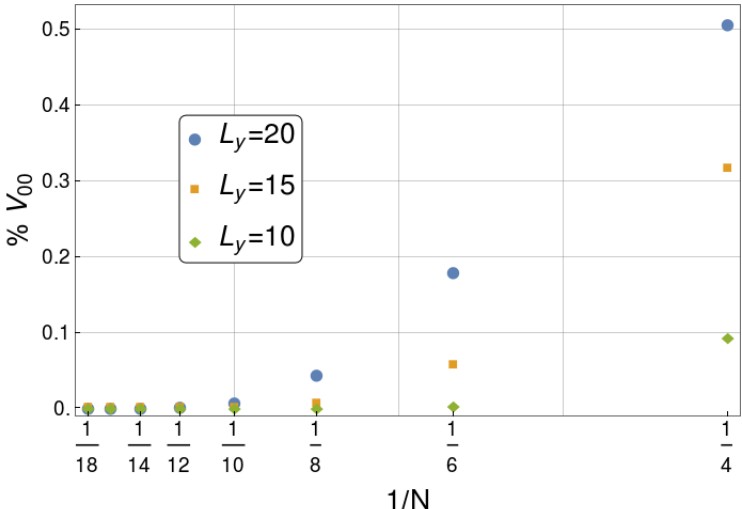

FIG. 9. The ratio of the smallest kept interaction matrix element $V_{k,m}$ (see Eq. (8)) to the largest, $V_{0,0}$, as function of total number of orbitals $N$. Interaction elements smaller than this are dropped. Magnetic length is $l_B = 1$.

### D. The small $l_s$ limit of the screened Coulomb repulsion

The screened Coulomb repulsion $V_{\rm sc}$ reduces to the contact interaction $V_\delta$ in the limit of small screening length $l_s$. By this we mean that both $l_s \ll L_y$ and $l_s \ll l_B$ hold. The soft exponential cutoff sharpens in this limit, and thus the interaction is only significant in a patch of size $l_s \ll L_y$, which is a tangent plane on the cylinder. Thus we write

Eq. (8) in polar coordinates defined by $\rho = \sqrt{\tilde{x}^2 + \tilde{y}^2} \in (0, l_s)$, and with angle $\theta \in (0, 2\pi)$. Eq. (8) takes the shape

$$V_{km}^{\text{sc}, l_s \to 0} = g \frac{\sqrt{\pi/2}}{L_y l_B} \int_0^{l_s} d\rho\, \rho \int_0^{2\pi} d\theta\, e^{-i \frac{2\pi k}{L_y} \rho \sin\theta}\, e^{-\frac{1}{2l_B^2}(\rho\cos\theta + l_B^2 \frac{2\pi m}{L_y})^2}\, e^{-\frac{1}{2} l_B^2 (\frac{2\pi k}{L_y})^2} \frac{e^{-\frac{\rho}{l_s}}}{\rho}.$$

Since $\sin\theta$ and $\cos\theta$ are non-singular for any $\theta$, as long as $k \ll L_y/l_s$, we can drop all terms of order $l_s/L_y$.

$$V_{km}^{\text{sc}, l_s \to 0} = 2\pi l_s (1 - e^{-1}) \left( g \frac{\sqrt{\pi/2}}{L_y l_B} e^{-\frac{1}{2} l_B^2 (\frac{2\pi m}{L_y})^2} e^{-\frac{1}{2} l_B^2 (\frac{2\pi k}{L_y})^2} \right), \tag{24}$$

where the term $l_s(1-e^{-1})$ comes from evaluating the average of the exponential cutoff over the patch, i.e $\int_0^{l_s} d\rho\, e^{-\rho/l_s} = l_s(1 - e^{-1})$. This is the same as the contact interaction Eq. (12) up to a coefficient. More specifically,

$$V_{km}^{\text{sc}, l_s \to 0} = 2\pi l_s (1 - e^{-1}) \frac{g}{g_\delta} V_{km}^\delta. \tag{25}$$

Therefore $g_\delta \propto l_s g$ in the limit of small $l_s$. For $g_\delta$ to be constant, we observe that $g \to \infty$ for contact-like interaction. This is reasonable, given that the strength of contact interaction (delta distribution function) diverges at the singular point. Because of this difference in units, one should not energetically compare Fig 3b and Fig 3a, which would lead one to believe that the gap for the contact interactions is smaller than for $l_s = 1 l_B$. In fact, shrinking the screening length increases the gap energy (as shown by Fig 10), even though the energies of the ground and excited states individually decrease with shrinking screening length.

If we instead consider the case of increasing screening length, the energy of the ground and first excited states increases, while their relative energy shrinks. As shown in Fig 10 for $l_s = 3 l_B$, there exists sufficiently large enough range of the interactions that the first excited states, which are edge states, come down in energy to form a new ground state for sufficiently large $N$. Despite this, the polarized states remain eigenstates at higher energy; and the new ground states, which do not have one-particle-per-site order, can be described as having the largest density at the center and edges. Because of these two facts, we expect that this breakdown in order is a consequence of the open boundary conditions. However peculiar, this new ground state (at $l_s = 3 l_B$) does not appear at small $N$, which indicates that the "blurring" of the interaction due to truncation (See Supplement C) is an unlikely suspect, as we expect that to decrease with increasing $N$. Nonetheless, since the new ground state at $l_s = 3 l_B$ is formed from edge states, which we see come down in relative energy to the ground state in Fig 10, and not the fully polarized states, it may be that this state would not exists without the presence of an edge. Infinite DMRG or a finite size system with a tunable confining edge potential, the latter which can push the edge states up in energy, are possible paths that future research could take in further investigating this.

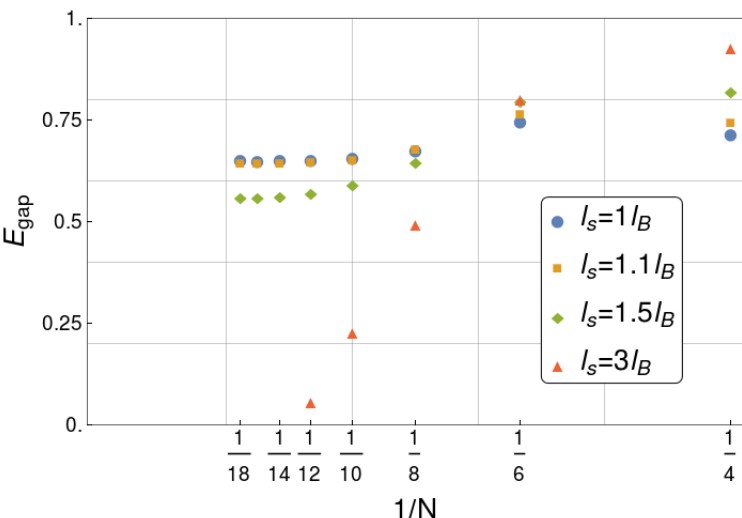

FIG. 10. The gap to the first excited states for $Z_2$ Hamiltonian and screened Coulomb interaction with different interaction range $l_s$, see legend with respect to total orbital number $N$ at $\gamma^{-1} = 10$. Increasing the interaction range decreases the gap and after some $l_s > 1.5 l_B$, the gap vanishes as the total orbital number $N$ increases, thus suggesting either inadequacy in numerics to capture the nature of the ground state or a change in the nature of the ground state for longer range interactions.

### E. Eigenstates and eigenenergies

Here we lay out the exact eigenenergies for the eigenstates $|++\cdots+\rangle$ and $|\psi\rangle$ as a function of orbital number $N$ and cylinder circumference $L_y$. Similarly to the main text, we notate our indices $n, k, m$, where $k, m$ are half-integers/integers for $N$ even/odd; and $n$ is always an integer. For the polarized state,

$$H_{\mathbb{Z}_2} |++\cdots+\rangle = \sum_n \theta_{(|n|\leq Q)} \sum_k \theta_{(|n+k|\leq Q)} \left( -V_{k,0} + V_{0,k} \right) |++\cdots+\rangle, \tag{26}$$

which is Eq. (10) with the bounds on the sums of $n$ and $k$ written explicitly as step functions. Likewise, for $|\psi\rangle = d^\dagger_{-,-Q} d_{+,Q} |++\cdots+\rangle$ (Eq. (13)):

$$H_{\mathbb{Z}_2} |\psi\rangle = \left( 2\left( \sum_{0 \leq m \leq 2Q} V_{0,m} \right) - 2V_{0,0} \right. \tag{27}$$

$$\left. + E_{|++\cdots+\rangle} - \left( \sum_{-2Q \leq k \leq 0} V_{0,k} - V_{k,0} \right) - \left( \sum_{0 \leq k \leq 2Q} V_{0,k} - V_{k,0} \right) \right) |\psi\rangle,$$

which, because of inversion symmetry of the 2-body interaction in Eq. (8), can be simplified to

$$= \left( 2\left( \sum_{0 < m \leq 2Q} V_{0,m} \right) + E_{|++\cdots+\rangle} - \left( \sum_{-2Q \leq k \leq 2Q} V_{0,k} - V_{k,0} \right) \right) |\psi\rangle.$$

The reader should be aware that these formula suffer from the same truncation error as discussed in the main text, even if the eigenstates themselves are unaffected, being product states independent of $N$. This truncation error manifests in the bounds on the indices, which throw out $V_{k,m}$'s outside those bounds.

### F. Two-point correlations in real space and their correlation holes

Throughout this paper, we point to the real-momentum space connection of the Landau-gauge LLL wavefunctions, which provides a useful guiding-center representation where we can understand the LLL wavefunctions as forming an effective 1D lattice (Fig 2). However, one needs to be careful in thinking of the guiding-center chain as a literal lattice in real space. This being because the chain is indexed by the eigenvalues of momentum about the cylinder circumference – not position. It is only as a convenient consequence of the non-trivial topology that, say, the localized LLL wavefunction center at $x = 0$ evolves into its neighbor's center at $x = k_y l_B^2$ when its momentum is increased adiabatically by $+k_y$. (And likewise, moving in the opposite direction $-k_y$ for the LLL wavefunctions which feel $B < 0$.) In truth, the LLL wavefunctions have non-zero amplitude everywhere on the cylinder (Eq. (5)), such that (for example) Eq. (13) may have non-zero energy despite it being a product state in the guiding-center representation. This peculiarity of the Landau gauge, when not taken carefully, can misinform intuitive thinking, and thus one may need to resort to the real-space picture in order to fully understand a state's energetics.

For example, in the presence of contact interactions, the fully polarized ground state $|++\cdots+\rangle$ has zero energy. This arising at the level of the projected interaction Eq. (8) as a symmetry of the matrix element $V_{km} = V_{mk}$, which generally forbids the like-Chern number electrons from interacting, and is additionally responsible for killing the energy of the polarized state. However, one might have immediately pointed out that such back flips are unnecessary, seeing that Pauli exclusion principle forbids same-flavour fermions from interacting at a point. Thus the fully polarized state, in spite of the wavefunctions overlapping in real space, must be at zero energy since it is composed of particles which are forbidden from interacting. More systematically, if we decompose the projected real-space annihilation operators into opposite-Chern components: $\hat{c}(\mathbf{r}) = \hat{c}_+(\mathbf{r}) + \hat{c}_-(\mathbf{r})$. The projected contact interaction reduces to a cross-term

$$= \sum_{\mathbf{r}} 2\hat{n}_+(\mathbf{r})\hat{n}_-(\mathbf{r}),$$

from which it follows that any state with only one flavour of fermion is a zero energy state.

This systematic approach can be expanded on by projecting the Hamiltonian onto the state

$$\langle ++\cdots+|H_{\mathbb{Z}_2}|++\cdots+\rangle = \tag{28}$$

$$\iint d\mathbf{r}\, d\mathbf{r}' \langle ++\cdots+| \hat{c}^\dagger(\mathbf{r})\hat{c}^\dagger(\mathbf{r}')\hat{c}(\mathbf{r}')\hat{c}(\mathbf{r}) |++\cdots+\rangle V(\mathbf{r}-\mathbf{r}'),$$

and studying how the various correlations weight the expectation value of the state. For the fully polarized state, we only need to calculate a single correlation function:

$$\rho_0(\mathbf{r}, \mathbf{r}') = \langle + + \cdots + | \hat{c}^\dagger(\mathbf{r}) \hat{c}^\dagger(\mathbf{r}') \hat{c}(\mathbf{r}') \hat{c}(\mathbf{r}) | + + \cdots + \rangle . \tag{29}$$

If, for a moment, we hold $\mathbf{r}'$ fixed, then one may interpret $\rho_0(\mathbf{r}, \mathbf{r}')$ as the conditional probability of finding an electron at $\mathbf{r}$ given an electron exists at fixed $\mathbf{r}'$. Now, if $\rho_0(\mathbf{r}, \mathbf{r}')$ is zero for some $\mathbf{r}$ and $\mathbf{r}'$, having a "correlation hole" for the position of those two particles, then no energy $V(\mathbf{r} - \mathbf{r}')$ is contributed to the overall energy of the state. For a contact interaction, this means the state is zero energy. Since $| + + \cdots + \rangle$ is in the guiding-center representation, we can use Eq. (16) to find a closed form expression for $\rho_0$. Doing so gives us (up to the normalization of the LLL wavefunction)

$$\rho(\mathbf{r}, \mathbf{r}') = \sum_{kk'pp'} \langle + + \cdots + | d^\dagger_{+,k} d^\dagger_{+,k'} d_{+,p'} d_{+,p} | + + \cdots + \rangle \tag{30}$$

$$\times\, e^{iy(-k_y + p_y)} e^{iy'(-k'_y + p'_y)} e^{-\frac{1}{2l^2}(x - l^2 k_y)^2} e^{-\frac{1}{2l^2}(x - l^2 p_y)^2} e^{-\frac{1}{2l^2}(x - l^2 k'_y)^2} e^{-\frac{1}{2l^2}(x - l^2 p'_y)^2},$$

where $k_y \equiv 2\pi k / L_y$. Writing $\langle \hat{\mathcal{O}} \rangle_0 \equiv \langle + + \cdots + | \hat{\mathcal{O}} | + + \cdots + \rangle$, this becomes

$$= \sum_{kk'pp'} \left( \langle d^\dagger_{+,k} d_{+,p} \rangle_0 \langle d^\dagger_{+,k'} d_{+,p'} \rangle_0 - \langle d^\dagger_{+,k} d_{+,p'} \rangle_0 \langle d^\dagger_{+,k'} d_{+,p} \rangle_0 \right)$$

$$\times\, e^{iy(-k_y + p_y)} e^{iy'(-k'_y + p'_y)} e^{-\frac{1}{2l^2}(x - l^2 k_y)^2} e^{-\frac{1}{2l^2}(x - l^2 p_y)^2} e^{-\frac{1}{2l^2}(x' - l^2 k'_y)^2} e^{-\frac{1}{2l^2}(x' - l^2 p'_y)^2},$$

and, using the constraints of quadratic correlation functions in the fully polarized state, further simplifies to

$$= \sum_{k,k'} e^{-\frac{1}{2l^2}(x - l^2 k_y)^2} e^{-\frac{1}{2l^2}(x' - l^2 k'_y)^2} \left( e^{-\frac{1}{2l^2}(x - l^2 k_y)^2} e^{-\frac{1}{2l^2}(x' - l^2 k'_y)^2} \right.$$

$$\left. -\, e^{i(y - y')(-k'_y + k_y)} e^{-\frac{1}{2l^2}(x' - l^2 k_y)^2} e^{-\frac{1}{2l^2}(x - l^2 k'_y)^2} \right).$$

Plotting this quantity (Fig 11) reveals a correlation hole for observing a $C = +1$ electron where a $C = +1$ electron already exists. Like-electrons in the presence of contact interactions never observe one another, forbidden by Pauli exclusion, which arises as the correlation hole, and thus a fully polarized state $| + + \cdots + \rangle$ is effectively non-interacting and zero-energy.

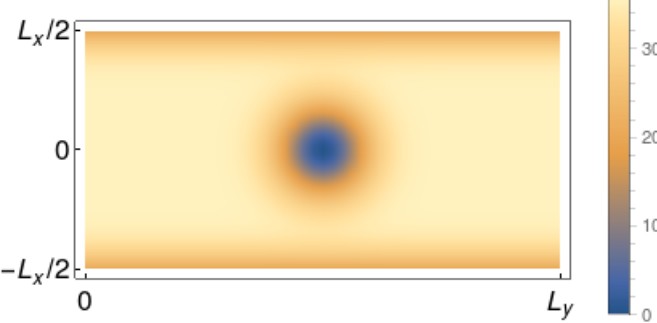

FIG. 11. $\rho_0(x, y, 0, \frac{L_y}{2})$, the ground state ($| + + \cdots + \rangle$) conditional probability density of finding a $C = +1$ electron somewhere given there is a $C = +1$ electron at position $(0, \frac{L_y}{2})$ on the cylinder. The correlation hole is isotropic. The (unrolled) cylinder periodicity in the y-direction is understood as matching $y = 0$ with $y = L_y$, and the number of available orbitals on the cylinder roughly lie in the segment between $(-\frac{L_x}{2}, \frac{L_x}{2})$, where $L_x = 2\pi N l_B^2 / L_y$. The system sizes here: $L_y = 15$, $l_B = 1$, and $N = 18$.

## G.    Understanding bilayer coherence: $H_{\frac{1}{2}\mathbf{QH}}$ vs $H_{\mathbb{Z}_2}$

Here we present a simple picture for understanding the layer coherence of the bilayer QH problem. Consider electrons living on either of two parallel plates separated by a distance $d$, and label the two layers by a pseudospin $\alpha \in \{\uparrow, \downarrow\}$, where $\uparrow(\downarrow)$ is the upper(lower) layer. If the plates are infinite in extent and approximately uniform

in charge with surface charge densities $\sigma_\alpha$, then translational symmetry exists such that for the potential of either plate $\phi_\alpha(x, y, z) = \phi_\alpha(z)$. Such a problem is effectively a one-dimensional point charge with charge replaced by the charge density: $-\partial_z \phi_\alpha(z) = \sigma_\alpha \delta(z - z_\alpha)$ for a plates at $z_\uparrow = +d/2$ and $z_\downarrow = -d/2$, which has the solution $\phi_\alpha(z) = -\frac{1}{2}\sigma_\alpha|z - z_\alpha|$. The electric field between the two plates is uniform and parallel (or anti-parallel) to the uniform magnetic field $\mathbf{B} = \hat{\mathbf{z}}B$. The electric charges $q_\downarrow$ in the lower plate experience the field $\phi_\uparrow$ of the upper plate (and visa-versa), leading to our approximation for the inter-layer interaction:

$$\hat{V}_{\text{inter}} = \int d\mathbf{r}\, dz\, d\mathbf{r}'\, dz'\, c_\uparrow^\dagger(\mathbf{r}, z) c_\downarrow^\dagger(\mathbf{r}', z') \Big(\frac{q_\downarrow \sigma_\uparrow |z - z'|}{-2}\Big) c_\downarrow(\mathbf{r}', z') c_\uparrow(\mathbf{r}, z), \tag{31}$$

which, for electrons confined to their respective layers ($z = z_\uparrow$ and $z' = z_\downarrow$) reduces to:

$$= \int d\mathbf{r}\, d\mathbf{r}'\, c_\uparrow^\dagger(\mathbf{r}) c_\downarrow^\dagger(\mathbf{r}') \Big(\frac{q_\downarrow \sigma_\uparrow d}{-2}\Big) c_\downarrow(\mathbf{r}') c_\uparrow(\mathbf{r})$$

or even more simply (defining the constant $\lambda \equiv q_\downarrow \sigma_\uparrow / 2$ and the total number operators $\hat{N}_\alpha$):

$$= -\lambda d\, \hat{N}_\uparrow \hat{N}_\downarrow.$$

Clearly, for a system with fixed particle number $\hat{N}_\alpha + \hat{N}_\beta = N$, the inter-layer interaction favours a state with $\hat{N}_+ = \hat{N}_- = N/2$. If we define pseudospin matrices $\tau_{\alpha\beta}^z$ to be the Pauli-$Z$ $SU(2)$ generator, we can rewrite the inter-layer interaction as a out-of-plane coupling of electronic pseudospins plus a density-density piece

$$\hat{V}_{\text{inter}} = \int d\mathbf{r}\, d\mathbf{r}'\, \frac{-\lambda d}{2} c_\alpha^\dagger(\mathbf{r}) c_\beta(\mathbf{r}) \Big( \mathbb{1}_{\alpha\beta} \mathbb{1}_{\alpha'\beta'} - \tau_{\alpha\beta}^z \tau_{\alpha'\beta'}^z \Big) c_{\alpha'}^\dagger(\mathbf{r}') c_{\beta'}(\mathbf{r}'). \tag{32}$$

When written in this way, it is clear the second $\tau^z \otimes \tau^z$ term breaks $SU(2)$ symmetry, and additionally disfavours states with an imbalance between the total number of particles between the two layers. If we consider this inter-layer potential in addition to the layer-separation-independent interactions $H_{\frac{1}{2}\text{QH}}$, where we project all single-particle operators onto the LLL by substituting $c_\alpha(\mathbf{r}) \to \hat{c}_\alpha(\mathbf{r})$, we obtain the total Hamiltonian

$$H = H_{\frac{1}{2}\text{QH}} + \hat{V}_{\text{inter}}. \tag{33}$$

For $d = 0$, the two layers are not separated, and $H$ reduces to $H_{\frac{1}{2}\text{QH}}$, which has a extensively degenerate ground state composed of polarized states and those connected to them by $SU(2)$ symmetry. Separating the layers ($d > 0$) by a small amount introduces anisotropy which splits the massively degenerate $SU(2)$ ground state manifold, driving up the polarized states in energy relative to those states with total $S_z = \int d\mathbf{r}\, c^\dagger(\mathbf{r}) \tau_z c(\mathbf{r}) = 0$.

For the scenario in which the two layers experience opposite sign magnetic fields ($H_{\mathbb{Z}_2}$), the ground state degeneracy is two-fold between fully polarized states, and there appears to exist a gap to the lowest excited states. Thus, no such instability exists.

## H.   1D Ising transverse field as wall defect in 2D

Let us start with the one-body potential

$$\sum_{\mathbf{r}} c_+^\dagger(\mathbf{r}) f(\mathbf{r}) c_-(\mathbf{r}) \tag{34}$$

plus its hermitian conjugate. The one-body potential term $f(\mathbf{r})$ is a wall defect at $y = y_w$ along the length of the cylinder of circumference $L_y$:

$$f(\mathbf{r}) = \delta(y - y_w), \tag{35}$$

which has the orbital representation

$$= \sum_q e^{iq(y - y_w)}.$$

Because of the cylinders axial symmetry, the position of the wall $y_w$ should be irrelevant. Projecting this onto the LLL (replacing $c_\pm(\mathbf{r}) \to \hat{c}_\pm(\mathbf{r})$ and plugging $f(\mathbf{r}) = \delta(y - y_w)$ into Eq. (34)) follows as

$$\sum_\mathbf{r} \hat{c}_+^\dagger(\mathbf{r}) f(\mathbf{r}) \hat{c}_-(\mathbf{r})$$

$$= \sum_\mathbf{r} \left( \sum_{k_y} e^{-ik_y y} e^{-\frac{1}{2l^2}(x - l_B^2 k_y)^2} d_{+,k_y}^\dagger \right) \delta(y - y_w) \left( \sum_{p_y} e^{ip_y y} e^{-\frac{1}{2l_B^2}(x + l_B^2 p_y)^2} d_{-,p_y} \right)$$

$$= \sum_{k_y, p_y} d_{+,k_y}^\dagger d_{-,p_y} \sum_y \delta(y - y_w) e^{-i(k_y - p_y)y} \sum_x e^{-\frac{1}{2l_B^2}(x - l_B^2 k_y)^2} e^{-\frac{1}{2l_B^2}(x + l_B^2 p_y)^2}.$$

Now using the Fourier transform of the delta-function $\delta(y - y_w) = \sum_q e^{iq(y - y_w)}$,

$$= \sum_{k_y, p_y} d_{+,k_y}^\dagger d_{-,p_y} \sum_y \left( \sum_q e^{iq(y - y_w)} \right) e^{-i(k_y - p_y)y} \sum_x e^{-\frac{1}{2l_B^2}(x - l_B^2 k_y)^2} e^{-\frac{1}{2l_B^2}(x + l_B^2 p_y)^2}$$

$$= \sum_{k_y, p_y, q} d_{+,k_y}^\dagger d_{-,p_y} e^{-iq y_w} \sum_y e^{-i(k_y - p_y - q)y} \sum_x e^{-\frac{1}{2l_B^2}(x - l_B^2 k_y)^2} e^{-\frac{1}{2l_B^2}(x + l_B^2 p_y)^2}$$

$$= \sum_{k_y, p_y, q} d_{+,k_y}^\dagger d_{-,p_y} e^{-iq y_w} \delta(p_y = k_y - q) \sum_x e^{-\frac{1}{2l_B^2}(x - l_B^2 k_y)^2} e^{-\frac{1}{2l_B^2}(x + l_B^2 p_y)^2}$$

$$= \sum_{k_y, q} d_{+,k_y}^\dagger d_{-,k_y - q} e^{-iq y_w} \sum_x e^{-\frac{1}{2l_B^2}(x - l_B^2 k_y)^2} e^{-\frac{1}{2l_B^2}(x + l_B^2(k_y - q))^2}.$$

Now note that the integral over $x$ is a convolution of two Gaussians, one centered at $x = l_B^2 k_y$ and the other at $x = l_B^2(k_y - q)$, with the largest contributions coming from those values of $q$ such that the Gaussians overlap, i.e $q = 2k_y$. The non-dominant terms' contribution depends on the ratio of the magnetic length to the cylinder circumference $l_B/L_y$:

$$= \sum_{k_y} d_{+,k_y}^\dagger d_{-,-k_y} e^{i2k_y y_w} + \text{non-dominant}.$$

The peculiar phase factor is a consequence of the choice of $y = 0$. This can be gauged away by redefining the plane wave part of Eq. (5). Thus we get

$$\sum_\mathbf{r} c_+^\dagger(\mathbf{r}) f(\mathbf{r}) c_-(\mathbf{r}) + h.c = \sum_{k_y} d_{+,k_y}^\dagger d_{-,-k_y} + d_{-,-k_y}^\dagger d_{+,k_y} + \text{non-dominant}. \tag{36}$$

Keeping only the dominant terms, we see that the largest effect of the wall defect is to mix the Chern, similar to the Ising tranverse field mixing of the spin, where the local Pauli-X is defined

$$X_n = \frac{1}{2} \left( d_{+,n}^\dagger d_{-,-n} + d_{-,-n}^\dagger d_{+,n} \right). \tag{37}$$