# Peer review of "DMRG study of strongly interacting $\mathbb{Z}_2$ flatbands: a toy model inspired by twisted bilayer graphene"

_SciPost Physics_

## Round 1 · Referee Report · Anonymous · 2020-6-13

Strengths

1. The given toy model for strongly interacting opposite-Chern flat bands is straightforward.
2. A connection between the excited states and 1D Ising-type spin physics is interesting.
3. A useful information about numerical approach to quantum Hall system is provided.

Weaknesses

1. The numerical results are rather obvious.
2. The discussion about numerical results is insufficient.
3. There is room for improvement on the presentation.

Report

The authors provided a simple model to capture the essential physics of two strongly interacting flat bands for the cases that the bands have identical (spin-1/2 QH) or opposite ($\mathbb{Z}_2$) Chern number. By studying the model numerically with a cylinder geometry, it was found that the gap to the first excited states remains finite for the $\mathbb{Z}_2$ case and it is significantly shrunk for the spin-1/2 QH case. Although the obtained results are rather obvious, the used numerical approaches would be useful information for future studies of quantum Hall system. Furthermore, the discussion on the nature of the excited states, especially a connection to 1D Ising-type spin physics, is interesting. Therefore, the contents of this manuscript seem to be enough for the publication in SciPost. However, I think that the discussion about numerical results is insufficient and ambiguous descriptions are seen in the presentation. Hence, I cannot recommend the publication in the present form. Then, I would like to request the improvement of the manuscript with addressing the following questions/suggestions.

- Is the number of LLL orbitals counted like $N=L_xL_y/(2\pi)$? If yes, how can one take $N=4, 6, 8,$ …, as shown in FIG.3 and FIG.4? Namely, what is the relation between $N$ and $L_x$?

- What is the definition of gap? Is it $\Delta E=E_1-E_0$? Also, does the single-particle gap provide the same result in the $\mathbb{Z}_2$ case or not? It would be a useful information for DMRG calculation.

- It is unclear how the parameters $l_s$, $l_B$ are chosen in FIG.3 and FIG.4. For example, the used choice of $l_B=1$ corresponds to $B=1$. Does it fulfill the condition that the gap between Landau levels is much larger than the interaction strength?

- In FIG.3, why the gap is larger for larger $L_y$, even though the gap between Landau levels typically depends only on $B$?

-Related to the above question, the data only for two circumferences may be not enough to confirm the saturation of the gap in the thermodynamic limit. If possible, it would be better to add one more data for another $L_y$ even if it is smaller than $L_y=15$.

- In FIG.3, one may naively think that the gap for the short-range contact-like interaction is larger than that for the screened Coulomb interaction but the result is opposite. Is the limit $l_s\to0$ of $V_{sc}$ quantitatively connected to $V_\delta$?

- Is the caption of FIG.4 correct? What does "plotted as $(\Delta E)^{−1}$" mean?

-The horizontal axis of FIG.4 should start from $1/N=0$. If possible, it would be better to discuss the gap in the thermodynamic limit.

- Which parameter value is used in FIG.5 and FIG.6?

- In FIG.5 and FIG. 6 the data for $L_y=2$ seems to deviate from the systematic size dependence. Is a periodic boundary indeed applied in the y-direction for $L_y=2$, namely, is the hopping taken twice?

- In FIG.7, it would be helpful for readers to put numbers in the color bar to estimate how strongly a spin is localized.

- The accuracy of DMRG should be written, even if the discarded weight is negligible.

Requested changes

1. The parameters and quantities should be clearly defined.
2. The reason why the used parameters were chosen should be clearly written.
2. More discussion about the interpretation of numerical results, especially for FIG.3 and FIG.4, should be added.
3. The presentation (including the caption) of some figures should be improved.

  • validity: good
  • significance: ok
  • originality: ok
  • clarity: ok
  • formatting: reasonable
  • grammar: acceptable

Author:  Paul Eugenio  on 2020-09-16  [id 972]

(in reply to Report 1 on 2020-06-13)

Please see the attached pdf.

Attachment:

sciPost_replyLetter.docx.pdf

---

## Editorial Decision

resubmitted